# DRoP: Distributionally Robust Data Pruning

**Artem Vysogorets**
Data Science Platform
Rockefeller University
amv458@nyu.edu

**Kartik Ahuja**
Meta FAIR

**Julia Kempe**
New York University
Meta FAIR

## ABSTRACT

In the era of exceptionally data-hungry models, careful selection of the training data is essential to mitigate the extensive costs of deep learning. Data pruning offers a solution by removing redundant or uninformative samples from the dataset, which yields faster convergence and improved neural scaling laws. However, little is known about its impact on classification bias of the trained models. We conduct the first systematic study of this effect and reveal that existing data pruning algorithms can produce highly biased classifiers. We present theoretical analysis of the classification risk in a mixture of Gaussians to argue that choosing appropriate class pruning ratios, coupled with random pruning within classes has potential to improve worst-class performance. We thus propose *DRoP*, a distributionally robust approach to pruning and empirically demonstrate its performance on standard computer vision benchmarks. In sharp contrast to existing algorithms, our proposed method continues improving distributional robustness at a tolerable drop of average performance as we prune more from the datasets.

## 1 INTRODUCTION

The ever-increasing state-of-the-art performance of deep learning models requires exponentially larger volumes of training data according to neural scaling laws (Hestness et al., 2017; Kaplan et al., 2020; Rosenfeld et al., 2020; Gordon et al., 2021). However, not all collected data is equally important for learning as it contains noisy, repetitive, or uninformative samples. A recent research thread on data pruning is concerned with removing those unnecessary data, resulting in improved convergence speed, scaling, and resource efficiency (Toneva et al., 2019; Paul et al., 2021; He et al., 2023). These methods design scoring mechanisms to assess the utility of each sample, often measured by its difficulty or uncertainty as approximated during a preliminary training round, that guides pruning. Sorscher et al. (2022) report that selecting high-quality data using these techniques can trace a Pareto optimal frontier, beating the notorious power scaling laws; Kolossov et al. (2024) demonstrate that data selection can improve training.

A recent study by (Pote et al., 2023) hints that data pruning may mitigate distributional bias in trained models, which is a well-established issue in AI systems concerning the performance disparity across classes (Dwork et al., 2012; Hardt et al., 2016b) or protected, minority groups of the population (e.g., race or gender). In Section 3, we conduct the first systematic evaluation of various pruning algorithms with respect to classification bias—a phenomenon characterized by highly disparate model performance across different classes—on a variety of benchmarks and conclude otherwise. For example, we find that Dynamic Uncertainty He et al. (2023) achieves superior average test performance with VGG-19 on CIFAR-100 and ResNet-50 on ImageNet but fails miserably in terms of worst-class accuracy. Thus, we argue that it is imperative to benchmark pruning algorithms using a more comprehensive suite of metrics that reflect classification bias, and to develop solutions that address distributional robustness directly.

To understand the fundamental principles that exacerbate classification bias for various pruning methods, we present a theoretical analysis of linear decision rules for a mixture of two isotropic Gaussians in Section 4. It also illustrates how simple random subsampling with difficulty-aware class-wise pruning quotas may yield better worst-class performance compared to existing, finer pruning algorithms that operate on a sample level. Based on these observations, we design a first "robustness-aware" data pruning protocol, coined Distributionally Robust Pruning (*DRoP*). It selects

target class proportions based on the corresponding class-wise error rates computed on a hold-out validation set after a preliminary training round on the full dataset. While these target quotas already improve robustness when combined with existing pruning algorithms, they particularly shine when applied together with random pruning in accordance with our theoretical analysis. Pruning with DRoP substantially reduces the classification bias of models even compared to training on the full dataset while offering enhanced data efficiency. We further verified the effectiveness of our method for initially imbalanced datasets and in the context of group robustness.

In hindsight, DRoP carries some analogies with distributionally robust optimization methods that address class/group bias, such as subsampling (Barua et al., 2012; Cui et al., 2019; Tan et al., 2020) or cost-sensitive learning that constructs error-based importance scores to reweigh the training loss, emphasizing data from the more difficult or minority classes (Sinha et al., 2022; Sagawa* et al., 2020; Liu et al., 2021; Idrissi et al., 2022; Wang et al., 2023; Lukasik et al., 2022; Chen et al., 2017). DRoP emulates this behavior with data pruning, retaining more samples from the more difficult classes. We discuss these similarities and compare DRoP to a prototypical cost-sensitive method, CDB-W (Sinha et al., 2022), on a variety of dataset types to further illustrate the effectiveness of our method.

The summary of our contributions and the structure of the remainder of the paper are as follows.

- In Section 3, using a standard computer vision testbed, we conduct the first comprehensive evaluation of existing data pruning algorithms through the lens of classification bias for a variety of datasets and architectures;

- In Section 4, we provide our theoretical analysis in the Gaussian mixture model to illustrate increased bias of current pruning methods, show how to optimize for worst-class risk and thus motivate our proposed solution;

- In Section 5, we propose a random pruning procedure with error-based class ratios coined *DRoP*, and verify its effectiveness in drastically reducing the classification bias. We also provide ablations to illustrate the strength of DRoP as a pruning method, which in some range is even able to improve over full-dataset cost-sensitive learning.

## 2 RELATED WORK

In the era where the training corpora of contemporary models are of web-scale size, improving data efficiency has become the focus of practitioners and researchers alike. The corresponding literature is exceptionally rich, with a few fruitful and relevant research threads. *Dataset distillation* replaces the original samples with synthetically generated ones that bear compressed, albeit not as much interpretable, training signal (Sucholutsky & Schonlau, 2021; Cazenavette et al., 2022; Such et al., 2020; Zhao & Bilen, 2023; Nguyen et al., 2021; Feng et al., 2024). *CoreSet methods* select representative samples that jointly capture the entire data manifold (Sener & Savarese, 2018; Guo et al., 2022; Zheng et al., 2023; Agarwal et al., 2020; Mirzasoleiman et al., 2020; Welling, 2009); they yield weak generalization guarantees for non-convex problems and are not too effective in practice, especially on larger datasets (Feldman, 2020; Paul et al., 2021). *Active learning* iteratively selects an informative subset of a larger pool of unlabeled data for annotation, which is ultimately used for supervised learning (Tharwat & Schenck, 2023; Ren et al., 2021; Beluch et al., 2018; Kirsch et al., 2019). *Subsampling* deletes instances of certain groups or classes when datasets are imbalanced, aiming to reduce bias and improve robustness of the downstream classifiers (Chawla et al., 2002; Barua et al., 2012; Chaudhuri et al., 2023).

**Data Pruning.** More recently, data pruning emerged as a new research direction that simply removes parts of the dataset while maintaining strong model performance. In contrast to previous techniques, data pruning selects a subset of the original, fully labeled, and not necessarily imbalanced dataset, all while enjoying strong results in deep learning applications. Data pruning algorithms use the entire dataset $\mathcal{D} = \{X_i, y_i\}_{i=1}^N$ to optimize a preliminary query model $\psi_\theta$ parameterized by $\theta$ that most often assigns "utility" scores $A(\psi, X)$ to each training sample $X$; then, the desired fraction $s$ of the least useful instances is pruned from the dataset, yielding a sparse subset $\mathcal{D}_s = \{X : A(\psi, X) \geq \text{quantile}[A(\psi, \mathcal{D}), s]\}$. In their seminal work, Toneva et al. (2019) let $A(\psi, X)$ be the number of times $(X, y)$ is both learned and forgotten while training the query model. Paul et al. (2021) design a "difficulty" measure $A(\psi, X) = \|\sigma[\psi(X)] - y\|_2$ where $\sigma$ denotes the softmax

function and $y$ is one-hot. These scores, coined EL2N, are designed to approximate the GraNd metric $A(\psi_\theta, X) = \|\nabla_\theta \mathcal{L}[\psi(X), y]\|_2$, which is simply the $\ell_2$-norm of the parameter gradient of the loss $\mathcal{L}$ computed at $(X, y)$. Both EL2N and GraNd scores require only a short training round (e.g., 10 epochs) of the query model. He et al. (2023) propose to select samples according to their dynamic uncertainty throughout training of the query model. For each training epoch $k$, they estimate the variance of the target probability $\sigma_y[\psi(X)]$ across a fixed window of $J$ previous epochs, and finally average those scores across $k$. Since CoreSet approaches are highly relevant for data pruning, we also consider one such label-agnostic procedure that greedily selects training samples that best jointly cover the data embeddings extracted from the penultimate layer of the trained query model $\psi$ (Sener & Savarese, 2018). While all these methods come from various contexts and with different motivations, several studies show that the scores computed by many of them exhibit high cross-correlation (Sorscher et al., 2022; Kwok et al., 2024).

**Robustness & Evaluation Metrics.**    Distributional robustness in machine learning concerns the issue of non-uniform accuracy over the data distribution (Sagawa* et al., 2020). The majority of the vast research in this area focuses on group robustness where certain, often under-represented or sensitive, groups of the population have worse predictive performance (Hashimoto et al., 2018; Thomas McCoy et al., 2020). In general, groups can be subsets of classes, and worst-group accuracy is a standard optimization criterion in this domain (Sagawa* et al., 2020; Kirichenko et al., 2023; Rudner et al., 2024). As a special case, fairness in machine learning aims to mitigate disparity of model performance across society and demographic groups and uses specific criteria such as equal opportunity and equalized odds (Hardt et al., 2016a; Caton & Haas, 2023). Many well-established algorithms in the space of distributional robustness apply importance weighting to effectively re-balance the long-tailed data distributions (cf. cost-sensitive learning (Elkan, 2001)). Thus, Sagawa* et al. (2020) optimize an approximate minimax *DRO (distributionally robust optimization)* objective using a weighted sum of group-wise losses, putting higher mass on high-loss groups; Sinha et al. (2022) weigh samples by the current class-wise misclassification rates measured on a holdout validation set; Liu et al. (2021) pre-train a reference model to estimate the importance factors of groups for subsequent re-training. Similar cost-weighting strategies are adopted for robust knowledge distillation (Wang et al., 2023; Lukasik et al., 2022) and online batch selection (Kawaguchi & Lu, 2020; Mindermann et al., 2022). Note that these techniques compute weights at the level of classes or sensitive groups and not for individual training samples. The focus of our study is a special case of group robustness, *classification bias*, where groups are directly defined by class attributions. Classification bias commonly arises in the context of imbalanced datasets where tail classes require upsampling or reweighting to produce models with strong worst-class performance (Cui et al., 2019; Barua et al., 2012; Tan et al., 2020; Chaudhuri et al., 2023). However, classification bias is also studied for balanced datasets, and is found to be exacerbated by adversarial training (Li & Liu, 2023; Benz et al., 2021; Xu et al., 2021; Nanda et al., 2021; Ma et al., 2022) and network *parameter* pruning (Paganini, 2020; Joseph et al., 2020; Tran et al., 2022; Good et al., 2022). Following mainstream prior work Li & Liu (2023); Zayed et al. (2023), we will use a natural suite of metrics to measure classification bias. Given accuracy (recall) $r_k$ for each class $k \in [K]$, we report (1) worst-class accuracy $\min_k r_k$, (2) difference between the maximum and the minimum recall, $\max_k r_k - \min_k r_k$ (Joseph et al., 2020), and (3) standard deviation of recalls $\text{std}_k r_k$ (Ma et al., 2022).

**Data Pruning Meets Robustness.**    Although existing data pruning techniques have proven to achieve strong average generalization performance, to our knowledge, our work contains the first comprehensive comparative study to call to the extensive worsening of classification bias of state-of-the-art data pruning methods. Among related works, Pote et al. (2023) studied how EL2N pruning affects the class-wise performance and found that, at high data density levels (e.g., 80–90% remaining data), under-performing classes improve their accuracy compared to training on the full dataset. Zayed et al. (2023) propose a modification of EL2N to achieve robustness across protected groups with two attributes (e.g., male and female) in datasets with counterfactual augmentation, which is a rather specific context. In this study, we eliminate this blind spot in the data pruning literature and analyze the trade-off between the average performance and classification bias exhibited by some of the most common data pruning methods: EL2N and GraNd (Paul et al., 2021), Forgetting (Toneva et al., 2019), Dynamic Uncertainty (He et al., 2023), and CoreSet (Sener & Savarese, 2018). We also benchmark random pruning (Random), which is regarded as a notoriously strong baseline in active learning and data pruning, especially when pruning large fractions of data.

# 3   DATA PRUNING IS NOT ROBUST

Figure 2 illustrates our evaluation of data pruning through the lens of class-wise robustness for two vision benchmarks, CIFAR-100 and TinyImageNet (additional plots for other metrics can be found in Appendix C). We present the experimental details and hyperparameters in Appendix A. First, we note that no pruning method is uniformly state-of-the-art, and the results vary considerably across model-dataset pairs. Among all algorithms, Dynamic Uncertainty with VGG-19 on CIFAR-100 and ResNet-50 on ImageNet presents a particularly interesting and instructive case. While it is arguably the best with respect to the average test performance, it fails miserably across all robustness metrics. Figure 1 left reveals that it actually removes entire classes already at $10\%$ of CIFAR-100. Therefore, it seems plausible that Dynamic Uncertainty simply sacrifices the most difficult classes to retain strong performance on the easier ones. Figure 16 confirms our hypothesis: in contrast to other algorithms, at low density levels,

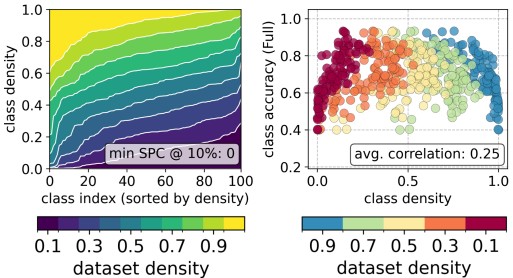

Figure 1: **Pruning Exacerbates Bias:** Dynamic Uncertainty applied to CIFAR-100. See Appendix G for similar plots for other pruning methods and models. **Left:** Sorted class densities at different dataset density levels. We also report the minimum number of samples per class (SPC) at $10\%$ dataset density. **Right:** Full dataset test class-wise accuracy against dataset density. We also report the correlation coefficient between these two quantities across classes, averaged over 5 dataset densities.

Dynamic Uncertainty tends to prune classes with lower baseline accuracy (obtained from training on the full dataset) more aggressively, which entails a catastrophic classification bias hidden underneath a deceptively strong average accuracy. This observation presents a compelling argument for using criteria beyond average test performance for data pruning algorithms, particularly emphasizing classification bias.

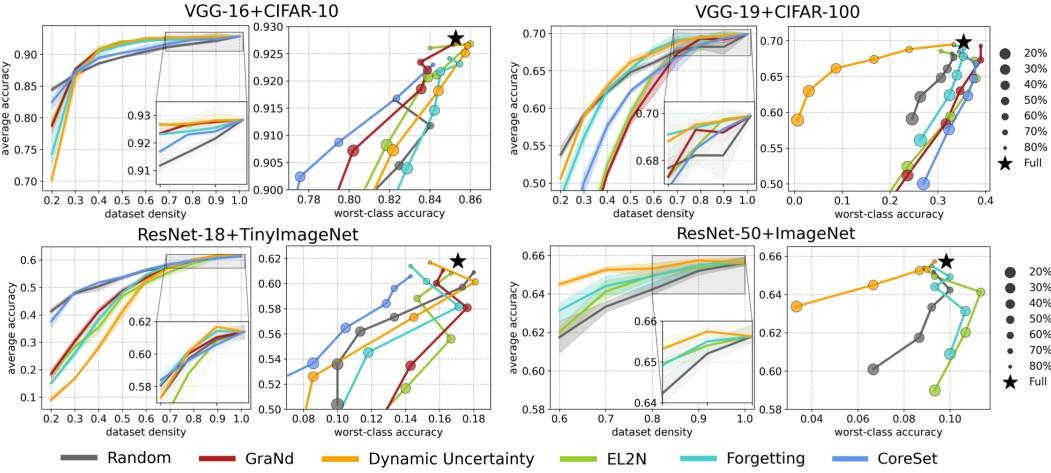

Figure 2: The average test performance of various data pruning algorithms against dataset density (fraction of samples remaining after pruning) and worst-class accuracy. All results averaged over 3 random seeds. Error bands represent min/max. Additional plots can be found in Appendix C.

Overall, all studied algorithms exhibit poor robustness to bias, although several of them improve ever so slightly over the full dataset. In particular, EL2N and GraNd achieve a relatively low classification bias, closely followed by Forgetting. At the same time, Forgetting has a substantially stronger average test accuracy compared to these three methods, falling short of Random only after pruning more than $60\%$ from CIFAR-100. As seen from additional plots in Appendix G, Forgetting produces more balanced datasets than EL2N and GraNd at low densities (Figure 17), and tends to prune "easier" classes more aggressively compared to all other methods (Figure 16). These two properties seem to be beneficial, especially when the available training data is scarce.

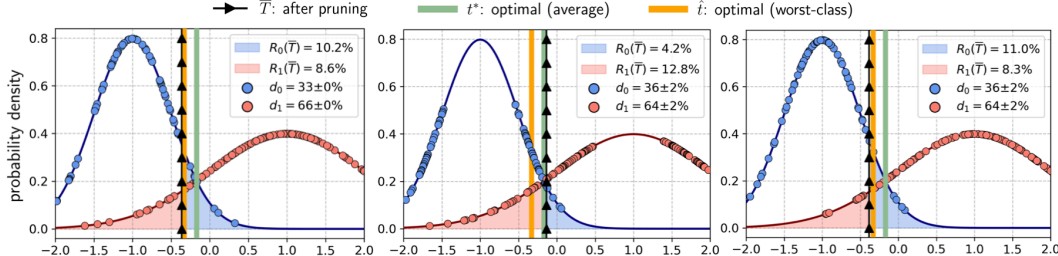

Figure 3: The effect of different pruning procedures on the solution mixture of Gaussians problem with $\mu_0 = -1$, $\mu_1 = 1$, $\sigma_0 = 0.5$, $\sigma_1 = 1$, and $\phi_0 = \phi_1$. Pruning to dataset density $d = 50\%$. **Left:** Random pruning with the optimal class-wise densities that satisfy $d_1\phi_1\sigma_0 = d_0\phi_0\sigma_1$. **Middle:** SSP. **Right:** Random pruning with respect to class ratios provided by the SSP algorithm. All results averaged across 10 datasets $\{D_i\}_{i=1}^{10}$ each with 400 points. The average ERM is $\overline{T} = \frac{1}{10}\sum_{i=1}^{10} T(D_i')$ fitted to pruned datasets $D_i'$. The class risks of the average and worst-class optimal decisions for this Gaussian mixture are $R_0[t^*(1)] = 4.8\%$, $R_1[t^*(1)] = 12.1\%$, and $R_0(\hat{t}) = R_1(\hat{t}) = 9.1\%$.

## 4    THEORETICAL ANALYSIS

In this section, we derive analytical results for data pruning in a toy model of binary classification for a mixture of two univariate Gaussians with linear classifiers. Perhaps surprisingly, in this framework we can derive worst-class optimal densities as well as demonstrate how prototypical pruning algorithms fail with respect to class robustness.

Let $\mathcal{M}$ be a mixture of two univariate Gaussians with conditional density functions $p(x|y = 0) = \mathcal{N}(\mu_0, \sigma_0^2)$ and $p(x|y = 1) = \mathcal{N}(\mu_1, \sigma_1^2)$, priors $\phi_0 = \mathbb{P}(y = 0)$ and $\phi_1 = \mathbb{P}(y = 1)$ ($\phi_0 + \phi_1 = 1$), and $\sigma_0 < \sigma_1$. Without loss of generality, we assume $\mu_0 < \mu_1$. Consider linear decision rules $t \in \mathbb{R} \cup \{\pm\infty\}$ with a prediction function $\hat{y}_t(x) = \mathbf{1}\{x > t\}$. The statistical 0-1 risks of the two classes are

$$R_0(t) = \Phi\left(\frac{\mu_0 - t}{\sigma_0}\right), \qquad R_1(t) = \Phi\left(\frac{t - \mu_1}{\sigma_1}\right), \tag{1}$$

where $\Phi$ is the standard normal cumulative distribution function. Under some reasonable but nuanced conditions on the means, variances, and priors discussed in Appendix B.1, the optimal decision rule minimizing the average risk

$$R(t) = \phi_0 R_0(t) + \phi_1 R_1(t) \tag{2}$$

is computed by taking the larger of the two solutions to a quadratic equation $\partial R / \partial t = 0$, which we denote by

$$t^*\left(\frac{\phi_0}{\phi_1}\right) = \frac{\mu_0\sigma_1^2 - \mu_1\sigma_0^2 + \sigma_0\sigma_1\sqrt{(\mu_0 - \mu_1)^2 - 2(\sigma_0^2 - \sigma_1^2)\log\frac{\phi_0\sigma_1}{\phi_1\sigma_0}}}{\sigma_1^2 - \sigma_0^2}. \tag{3}$$

Note that $t^*(\phi_0/\phi_1)$ is the rightmost intersection point of $\phi_0 f_0(t)$ and $\phi_1 f_1(t)$ where $f_0$ and $f_1$ are the corresponding probability density functions (Cavalli, 1945). Note that in the balanced case ($\phi_0 = \phi_1$) the heavier-tailed class is more difficult and $R_1[t^*(1)] > R_0[t^*(1)]$.

We now turn to the standard (class-) distributionally robust objective: minimizing worst-class statistical risk gives rise to the decision threshold denoted by $\hat{t} = \operatorname{argmin}_t \max\{R_0(t), R_1(t)\}$. As we show in Appendix B.2, $\hat{t}$ satisfies $R_0(\hat{t}) = R_1(\hat{t})$, and Equation 1 then immediately yields

$$\hat{t} = (\mu_0\sigma_1 + \mu_1\sigma_0)/(\sigma_0 + \sigma_1). \tag{4}$$

Note that $\hat{t} < t^*(1)$. This means that in the balanced case $\hat{t}$ is closer to $\mu_0$, the mean of the "easier" class. To understand how we should prune to achieve optimal worst-class accuracy, we compute class priors $\phi_0$ and $\phi_1$ that guarantee that the average risk minimization (Equation 2) achieves the best worst-class risk. These priors can be seen as the optimal class proportions in a "robustness-aware" dataset. Observe that, from Equation 3, $t^*(\sigma_0/\sigma_1) = \hat{t}$ because the logarithm in the discriminant

vanishes and we obtain Equation 4. Therefore, the optimal average risk minimizer coincides with the solution to the worst-case error over classes, and is achieved when the class priors $\phi_0, \phi_1$ satisfy

$$\phi_0/\phi_1 = \sigma_0/\sigma_1. \tag{5}$$

Intuitively, sufficiently large datasets sampled from $\mathcal{M}$ with class-conditionally independent samples mixed in proportions $N_0$ and $N_1$ should have their empirical 0-1 risk minimizers close to $t^*(N_0/N_1)$. That is, randomly retaining $d_k$ fraction of samples in class $k = 0, 1$ where $d_0 N_0 \sigma_1 = d_1 N_1 \sigma_0$ (see Equation 5) yields robust solutions with equal statistical risk across classes and, hence, optimal worst-class error. Figure 3 (left) confirms our analysis: the average empirical risk minimizer (ERM) $\overline{T}$ fitted to datasets pruned in this fashion lands near $\hat{t}$.

The class-conditional independence assumption we made above is crucial. While it is respected when subsampling randomly within each class, it clearly does not hold for existing, more sophisticated, data pruning algorithms. Therefore, even though they tend to prune easier classes more aggressively as evident from Figure 16, they rarely enjoy any improvement of the worst-class performance compared to the original dataset. We further illustrate this observation by replicating a supervised variant of the Self-Supervised Pruning (SSP) developed by Sorscher et al. (2022) for ImageNet. In this algorithm, we remove samples globally (i.e., irrespective of their class membership) located within a certain margin $M > 0$ of their class means. Intuitively, this algorithm discards the easiest or the most representative samples from the most densely populated regions of the distribution. As illustrated in Figure 3 (middle), this method fortuitously prunes the easier class more aggressively; indeed, the amount of samples removed from a class with mean $\mu$ and variance $\sigma^2$ is proportional to the area under the probability density function over the pruning interval $[\mu - M, \mu + M]$, which is larger for smaller values of $\sigma$. Nevertheless, if the original dataset has classes mixed in proportions $N_0$ and $N_1$, the solution remains close to $t^*(N_0/N_1)$ even after pruning, as we show formally in Appendix B.3. On the other hand, random subsampling according to class-wise pruning proportions defined by SSP does improve worst-class accuracy, as illustrated in the right plots of Figure 3. This corresponds to our observation in Figure 5 that random pruning respecting the class proportions discovered by GraNd and Forgetting often improves robustness compared to these methods themselves.

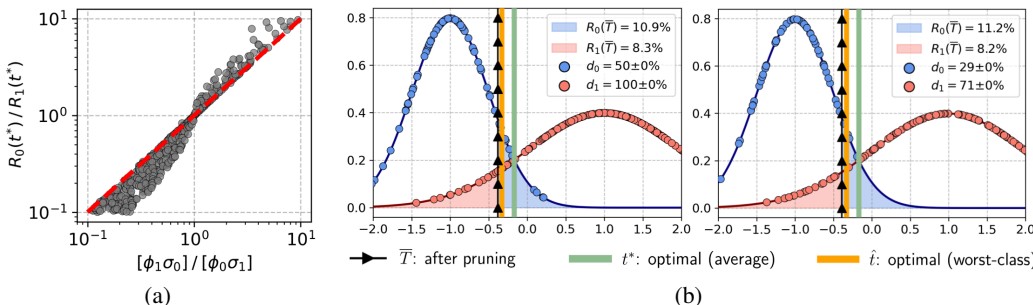

Figure 4: **(a)**: Class-wise risk ratios of the optimal solution $t^* = t^*(\phi_0/\phi_1)$ vs. optimal ratios based on Equation 5 computed for various $\sigma_0 < \sigma_1$ drawn uniformly from $[10^{-2}, 10^2]$ and $\phi_0 \sim U[0, 1]$ and $\phi_1 = 1 - \phi_0$. The results are independent of $\mu_0, \mu_1$. **(b)**: Random pruning with DRoP. **Left:** $d = 75\%$; **Right:** $d = 50\%$.

A priori, it is unclear how exactly the variance-based worst-case optimal pruning quotas in Equation 5 generalize to deep learning, and in particular how they can lead to practical pruning algorithms. One could connect our theory to the feature distribution in the penultimate layer of neural networks and determine class densities from cluster variances around class means. In fact, SSP (Sorscher et al., 2022) uses such metrics to determine pruning scores for samples, using a pretrained model. However, such variances are noisy, especially for smaller class sizes, and hard to connect to class accuracies. Therefore, instead, we propose to use *class errors* as a proxy for these variances in our DRoP method introduced in Section 5. More formally, given a dataset originally with class sizes $N_0$ and $N_1$, we replace the worst-class optimal condition $d_0 N_0 \sigma_1 = d_1 N_1 \sigma_0$ with $d_0 R_1[t^*(N_0/N_1)] = d_1 R_0[t^*(N_0/N_1)]$. To motivate this practical approach, Figure 4a shows that these new error-based class densities approximate the optimal variance-based ones fairly well, especially when $\sigma_1/\sigma_0$ is small. Figure 4b demonstrates that random pruning according to DRoP class proportions lands the average ERM near the worst-class optimal value. Thus, even though error-based class quotas do not enjoy simple theoretical guarantees, they still operate near-optimally in this toy setup.

## 5 DRoP: DISTRIBUTIONALLY ROBUST PRUNING

We are ready to propose our "robustness-aware" data pruning method, which consists in random subsampling according to carefully selected target class-wise sizes, incorporating lessons learned from our theoretical analysis. We propose to select the pruning fraction of each class based on its validation performance given a preliminary model $\psi$ trained on the whole dataset. Such a query model is still required by all existing data pruning algorithms to compute scores, so we introduce no additional resource overhead (see Appendix E for more discussion).

Consider pruning a $K$-way classification dataset originally with $N$ samples down to density $0 \le d \le 1$, so the target dataset size is $dN$ (prior literature sometimes refers to $1 - d$ as the pruning fraction). Likewise, for each class $k \in [K]$, define $N_k$ to be the original number of samples so $N = \sum_{k=1}^{K} N_k$, and let $0 < d_k \le 1$ be the desired density of that class after pruning. Then, we set $d_k \propto 1 - r_k$ where $r_k$ denotes recall (accuracy) of class $k$ computed by $\psi$ on a hold-out validation set. In particular, we define *DRoP quotas* as $d_k = d(1 - r_k)/Z$ where $Z = \sum_{k=1}^{K} (1 - r_k)N_k/N$ is a normalizing factor to ensure that the target density is respected, i.e., $dN = \sum_{k=1}^{K} d_k N_k$. Alas, not all dataset densities $d \in [0, 1]$ can be associated with a valid DRoP collection; indeed, for large enough $d$, the required class proportions may demand $d_k > 1$ for some $k \in [K]$. In such a case, we do not prune such classes and redistribute the excess density across unsaturated ($d_k < 1$) classes according to their DRoP proportions. The full procedure is described in Algorithm 1. This algorithm is guaranteed to terminate as the excess decreases in each iteration of the outer loop.

---

**Algorithm 1:** DRoP

**Input:** Target dataset density $d \in [0, 1]$.
  For each class $k \in [K]$: original size $N_k$, validation recall $r_k \in [0, 1]$.

**Initialize:** Unsaturated set of classes
  $U \leftarrow [K]$, excess $E \leftarrow dN$, class densities $d_k \leftarrow 0 \ \forall k \in [K]$.

**while** $E > 0$ **do**
  $Z \leftarrow \frac{1}{E} \sum_{k \in U} N_k(1 - r_k)$;
  **for** $k \in U$ **do**
    $d'_k \leftarrow (1 - r_k)/Z$;
    $d_k \leftarrow d_k + d'_k$;
    $E \leftarrow E - N_k d'_k$;
    **if** $d_k > 1$ **then**
      $U \leftarrow U \setminus \{k\}$;
      $E \leftarrow E + N_k(d_k - 1)$;
      $d_k \leftarrow 1$
    **end**
  **end**
**end**

**Return :** $\{d_k\}_{k=1}^{K}$.

---

**Evaluation.** To validate the effectiveness of random pruning with DRoP (Random+DRoP) in reducing classification bias, we compare it to various baselines derived from the strongest pruning algorithms: EL2N (Paul et al., 2021) (for CIFAR-10 and ImageNet), GraNd (Paul et al., 2021) and Forgetting (Toneva et al., 2019) (for CIFAR-100 and TinyImageNet). In addition to plain random pruning (Random)—removing a random subset of all training samples—for each of these two strategies, we consider (1) random pruning that respects class-wise ratios automatically determined by the strategy (Random+StrategyQ), (2) applying the strategy for pruning within classes but distributing sample quotas across classes according to DRoP (Strategy+DRoP), and (3) the strategy itself. The motivation for (1) is the anecdotal observation that existing pruning algorithms automatically balance classes according to their validation errors fairly well (Figure 16), while their overly scrupulous filtering within classes may be suboptimal. In support of this view, Ayed & Hayou (2023) formally show that integrating random sampling into score-based pruning procedures improves their average performance. Next, (2) serves as a reasonable ablation, while (3) is the benchmark we compare to. The implementation details are listed in Appendix A.

**Results.** Figure 5 presents our empirical results (additional plots showcasing other robustness metrics can be found in Appendix D). Overall, DRoP with random pruning consistently exhibits a significant improvement in distributional robustness of the trained models. In contrast to all other baselines that arguably achieve their highest worst-class accuracy at high dataset densities (80–90%), our method reduces classification bias induced by the datasets as pruning continues, e.g., up to 30–40% dataset density of TinyImageNet. Notably, Random+DRoP improves all robustness metrics compared to the full dataset on all model-dataset pairs, offering both robustness and data efficiency at the same time. For example, when pruning half of CIFAR-100, we achieve an increase in the worst-class accuracy of VGG-19 from 35.8% to 45.4%—an almost 10% change at a price of under 6% of the average performance. The leftmost plots in Figure 5 reveal that Random+DRoP does suffer a slightly larger degradation of the average accuracy as dataset density decreases compared to global

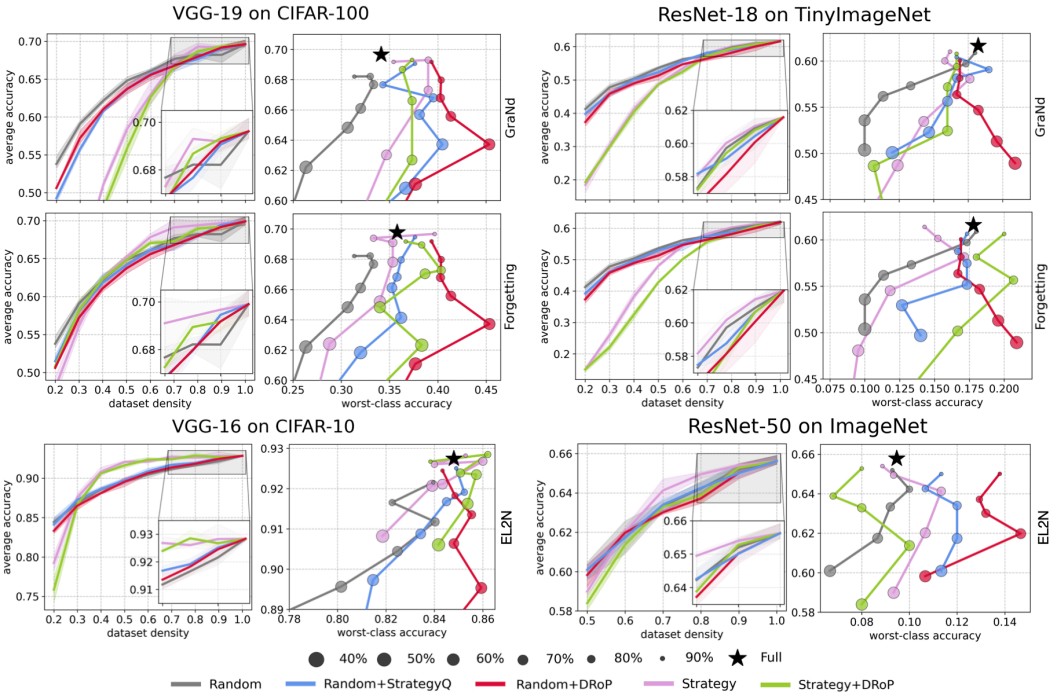

Figure 5: The average test performance of various data pruning protocols against dataset density and worst-class accuracy. All results averaged over 3 random seeds. Error bands represent min/max.

random pruning, which is unavoidable given the natural trade-off between robustness and average performance. Yet at these low densities, the average accuracy of Random+DRoP exceeds that of all other pruning algorithms.

As demonstrated in Figure 6, DRoP produces exceptionally imbalanced datasets unless the density $d$ is too low by heavily pruning easy classes while leaving the more difficult ones intact. As expected from its design, the negative correlation between the full-dataset accuracy and density of each class is much more pronounced for DRoP compared to existing pruning methods (cf. Figure 16).

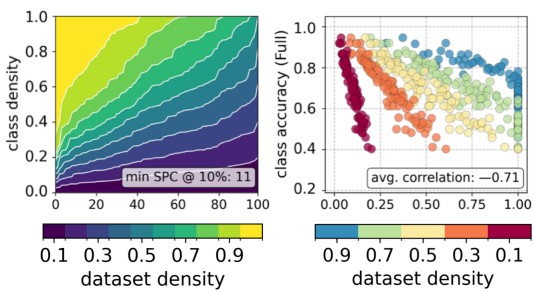

Figure 6: DRoP. **Left:** Sorted class densities at different dataset density levels. We report the minimum number of samples per class (SPC) at $10\%$ dataset density. **Right:** Full dataset test class-wise accuracy against dataset density. We also report the correlation coefficient between these two quantities across classes, averaged over 5 dataset densities.

Based on our examination of these methods in Section 3, we conjectured that these two properties are associated with smaller classification bias, which is well-supported by DRoP. Not only does it achieve unparalleled performance with random pruning, but it also enhances robustness of GraNd and Forgetting: Strategy+DRoP curves often trace a much better trade-off between the average and worst-class accuracies than their original counterparts (Strategy). At the same time, Random+StrategyQ fares similarly well, surpassing the vanilla algorithms, too. This indicates that robustness is achieved not only from the appropriate class ratios but also from pruning randomly as opposed to cherry-picking hard samples.

**A DRO baseline.** The results in Figure 5 provide solid evidence that Random+DRoP is by far the state-of-the-art data pruning algorithm in the robustness framework. Not only is it superior to prior pruning baselines, it in fact produces significantly more robust models compared to the full dataset, too. Thus, we go further and test our method against one representative cost-sensitive learning method from the DRO literature, Class-wise Difficulty Based Weighted loss (CDB-W) (Sinha et al., 2022). In its simplest form adopted in this study, CDB-W dynamically updates class-specific weights $w_{k,t} = 1 - r_{k,t}$ at every epoch $t$ by computing recalls $r_{k,t}$ on a holdout validation set, which is

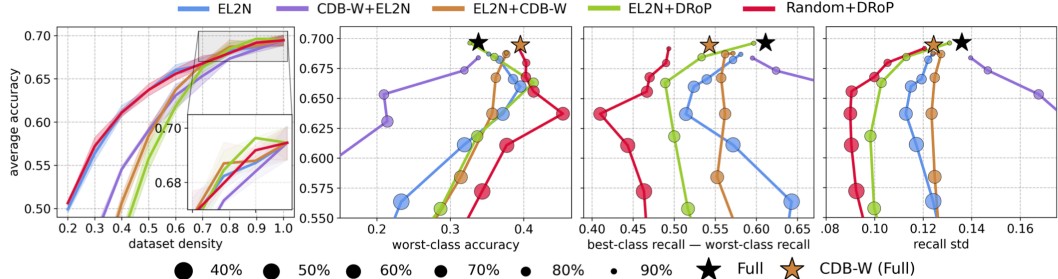

Figure 7: The average test performance of data pruning protocols against data density and measures of class robustness (VGG-19 on CIFAR-100). All results averaged over 3 random seeds. Error bands represent min/max.

precisely the proportions used by DRoP. Throughout training, CDB-W uses these weights to upweigh the per-sample losses based on the corresponding class labels. As part of the evaluation, we test if the robustness-driven optimization of CDB-W can help data pruning (for these experiments, we use EL2N). To this end, we consider two scenarios: training the final model with CDB-W on the EL2N-pruned dataset (EL2N+CDB-W), and using a robust query model trained by CDB-W to generate EL2N scores (CDB-W+EL2N).

Having access to the full dataset for training, CDB-W improves the worst-class accuracy of VGG-19 on CIFAR-100 by $5.7\%$ compared to standard optimization, which is almost twice as short of the increase obtained by removing $50\%$ of the dataset with Random+DRoP (Figure 5). When applied to EL2N-pruned datasets, CDB-W maintains that original bias across sparsities, which is clearly inferior not only to Random+DRoP but also to EL2N+DRoP. Perhaps surprisingly, EL2N with scores computed by a query model trained with CDB-W fails spectacularly, inducing one of the worst bias observed in this study. Thus, DRoP can compete with other existing methods that directly optimize for worst-class accuracy.

**Imbalanced datasets.** In the literature, robustness almost exclusively arises in the context of long-tailed distributions where certain classes or groups appear far less often than others; for example, CDB-W was evaluated in this setting. While dataset imbalance may indeed exacerbate implicit bias of the trained models towards more prevalent classes, our study demonstrates that the key to robustness lies in the appropriate, difficulty-based class proportions rather than class balance per se. Even though the overall dataset size decreases, pruning with DRoP can produce far more robust models compared to full but balanced datasets (Figure 6). Still, to promote consistency in evaluation and to further validate our algorithm, we consider long-tailed classification scenarios. We follow the approach used by Cui et al. (2019) to inject imbalance into the originally balanced TinyImageNet. In particular, we subsample each class $k \in [K]$ and retain $\mu^{k-1}$ of its original size for some $\mu \in (0, 1)$. For an initially balanced dataset, the size ratio between the largest ($k = 1$) and the smallest ($k = K$) classes then becomes $\mu^{1-K}$, which is called the *imbalance factor* denoted by $I$. Figure 8 reveals that Random+DRoP consistently beats EL2N in terms of both average and robust performance across a range of imbalance factors ($I = 2, 5, 20$). Likewise, it almost always reduces bias of the unpruned imbalanced TinyImageNet even when training with a robustness-aware CDB-W procedure.

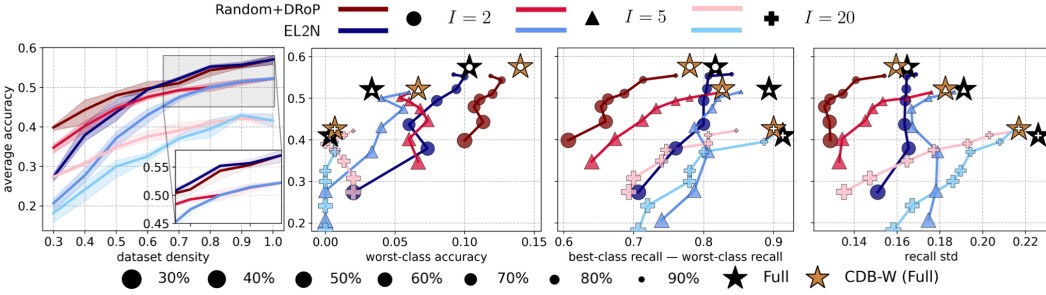

Figure 8: The average test performance of Random+DRoP (red-toned curves) and EL2N (blue-toned curves) against dataset density and measures of class robustness across dataset imbalance factors $I = 2, 5, 20$. ResNet-18 on imbalanced TinyImageNet. Results averaged over 3 random seeds. Error bands represent min/max.

**Group Distributional Robustness: Waterbirds.** While we scoped our study around robustness to classification bias, a vast majority of research is concerned with strong model performance across sensitive or minority groups within the distribution. One common benchmark dataset in this area is Waterbirds, which is a binary image classification of bird types where ground truth is spuriously correlated with image background (forest/water) for the majority of samples Sagawa* et al. (2020). Thus, standard optimization techniques produce models that often make predictions based on the background type rather than bird features, yielding poor accuracy for images from waterbird+forest or landbird+water groups. We use the original learning setup introduced by Sagawa* et al. (2020) to evaluate DRoP against other pruning baselines as well as CDB-W. Figure 9 shows that Random+DRoP and EL2N+DRoP achieve a significant improvement of the worst-group accuracy and other metrics. For worst-class accuracy, we adopt a handful of prior art including Just Train Twice (Liu et al., 2021), Learning from Failure (Nam et al., 2020), Learning to Split (Bao & Barzilay, 2022), and Bias Amplification (Li et al., 2023); the performance of these algorithms in this setup is reported by Pezeshki et al. (2024), which we copied verbatim. As evident from Figure 9, Random+DroP is well on par with these more sophisticated techniques that operate on full datasets. Therefore, as expected, our algorithm applies not only to reduce classification bias but also to improve group-wise robustness.

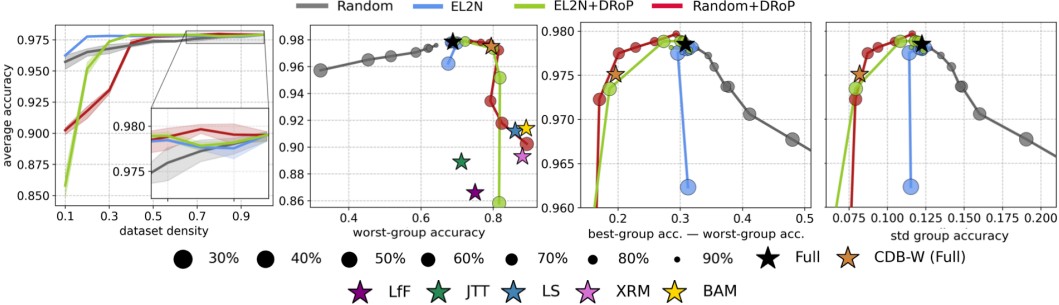

Figure 9: The average test performance of data pruning protocols and existing baselines against measures of group-wise robustness (ResNet-50 on Waterbirds). The results of data pruning and CDB-W averaged over 3 random seeds. Error bands represent min/max. To conform with Sagawa* et al. (2020), for this dataset, we compute average accuracy as a sum of group accuracies weighted by the original training group proportions. This explains the sharp degradation of the average performance of DRoP-backed pruning at low densities ($d \leq 0.4$): these datasets are skewed towards minority groups that weigh much less than severely pruned majority groups. Please see Appendix F for an extended discussion.

## 6 DISCUSSION

Data pruning—removal of uninformative samples from the training dataset—offers much needed efficiency in deep learning. However, all existing pruning algorithms are currently evaluated exclusively on their average performance, ignoring their potentially disparate impact on model predictions across data distribution. Through a systematic study of the classification bias, we reveal that current methods often exacerbate the performance disparity across classes, which can deceptively co-occur with high average performance. This leads us to formulate error-based class-wise pruning quotas coined DRoP. At the same time, we find value in pruning randomly within classes, as opposed to cherry-picking individual samples, which is inherent to the existing data pruning techniques. We confirm the effectiveness of our method on a series of standard computer vision benchmarks; our simple pruning protocol traces the best trade-off between average and worst-class performance among all existing data pruning algorithms and related baselines. Additionally, we find theoretical justification for the phenomenal success of this simple strategy in a toy classification model.

**Limitations & Future Work.** In this study, we focused our empirical evaluation primarily on classification bias. Thus, we only scratched the surface of robustness in deep learning, which is often concerned with group-wise model performance. Further research may attempt to understand the effect of DRoP and data pruning at large on worst-group accuracy and spurious correlations more deeply. Finally, we attribute our contributions mostly to research on data pruning and, therefore, present limited cross-evaluation with a broad spectrum of distributionally robust optimization methods. Thus, future research is warranted to perform a more detailed comparative study between these approaches.

## ACKNOWLEDGEMENTS

AV and JK acknowledge support through the NSF under award 1922658. Part of this work was done while JK was visiting the Centre Sciences de Donnees (CSD) at the Ecole Normale Superieure in Paris, France, and JK wishes to thank the CSD for their hospitality. This work was supported through the NYU IT High Performance Computing (HPC) resources, services, and staff expertise.

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

# A    Implementation Details

Our empirical work encompasses three standard computer vision benchmarks (Table 1). All code is implemented in PyTorch (Paszke et al., 2017) and run on an internal cluster equipped with NVIDIA RTX8000 GPUs. The total runtime of the empirical work presented in this paper is approximately $45,000$ GPU hours. We make our code available at `https://github.com/avysogorets/drop-data-pruning`.

**Data Pruning.**    Data pruning methods require different procedures for training the query model and extracting scores for the training data. For EL2N and GraNd, we use $10\%$ of the full training length reported in Table 1 before calculating the importance scores, which is more than the minimum of 10 epochs recommended by Paul et al. (2021). To improve the score estimates, we repeat the procedure across 5 random seeds and average the scores before pruning. Forgetting and Dynamic Uncertainty operate during training, so we execute a full optimization cycle of the query model but only do so once. Likewise, CoreSet is applied once on the fully trained embeddings. We use the greedy k-center variant of CoreSet. Since some of the methods require a hold-out validation set (e.g., DRoP, CDB-W), we reserve $50\%$ of the test set for this purpose. This split is never used when reporting the final model performance.

**Data Augmentation.**    We employ data augmentation only when optimizing the final model. The same augmentation strategy is used for all datasets except for Waterbirds where we used none. In particular, we normalize examples per-channel and randomly apply shifts by at most 4 pixels in any direction and horizontal flips.

**Models & Datasets.**    As shown in Table 1, we use the following model-dataset pairs: VGG-16 and VGG-19 (Simonyan & Zisserman, 2015) on CIFAR-10 and CIFAR-100 (Krizhevsky, 2009), respectively, ResNet-18 (He et al., 2016) on TinyImageNet (MIT License) (Le & Yang, 2015), ImageNet pre-trained ResNet-50 on Waterbirds (Sagawa* et al., 2020) (MIT License), and ResNet-50 on ImageNet (Deng et al., 2009). We also use a slight adaptation of Wide-ResNet-101 (Zagoruyko, 2016) on CIFAR-100 with a few downsampling layers removed.

| Model | Dataset | Epochs | Drop Epochs | Batch | LR | Decay |
|-------|---------|--------|-------------|-------|-----|-------|
| VGG-16 | CIFAR-10 | 160 | 80/120 | 128 | 0.1 | $1e\text{-}4$ |
| VGG-19 | CIFAR-100 | 160 | 80/120 | 128 | 0.1 | $5e\text{-}4$ |
| ResNet-18 | TinyImageNet | 200 | 100/150 | 256 | 0.2 | $1e\text{-}4$ |
| ResNet-50 | ImageNet | 90 | 30/60 | 512 | 0.2 | $1e\text{-}4$ |
| ResNet-50 | Waterbirds | 300 | None | 128 | 0.0001 | $1e\text{-}4$ |
| Wide-ResNet-101 | CIFAR-100 | 200 | 60/120/160 | 128 | 0.1 | $1e\text{-}4$ |

Table 1: Summary of experimental work and hyperparameters. All architectures include batch normalization (Ioffe & Szegedy, 2015) layers followed by ReLU activations. Models are initialized with Kaiming normal (He et al., 2015) and optimized by SGD (momentum 0.9) with a stepwise LR schedule ($0.2\times$ drop factor applied on specified Drop Epochs) and categorical cross-entropy. These hyperparameters are adopted from prior studies (Frankle et al., 2021; Wang et al., 2020; Sagawa* et al., 2020; Cai et al., 2022).

# B    Theoretical Analysis for a Mixture of Gaussians

Consider the Gaussian mixture model and the hypothesis class of linear decision rules introduced in Section 4. Here we give a more formal treatment of the assumptions and claims made in that section.

## B.1    Average risk minimization

Recall that we consider the average risk of the linear decision $\hat{y}_t(x) = \mathbf{1}\{x > t\}$ as $R(t) = \mathbb{E}_{x,y}[\ell(\hat{y}_t(x), y)]$, where the expectation is over $(x, y) \sim p(x, y)$ and class-conditional risk as

$R_y(t) = \mathbb{E}_{x|y}[\ell(\hat{y}_t(x)), y)]$, where the expectation is over $x \sim p(x|y)$ for $y \in \{0, 1\}$. If $\ell$ is the 0-1 loss, we thus obtain Equations 1 and 2. Recall that we have assumed $\sigma_0 < \sigma_1$ and $\mu_0 < \mu_1$.

We now give the precise conditions under which the average risk minimizer takes the form of $t^*(\phi_0/\phi_1)$ from Equation 3—the larger intersection point between the graphs of scaled probability density functions $\phi_0 f_0(t)$ and $\phi_1 f_1(t)$.

First, we make an assumption that this intersection exists, i.e., the expression under the square root in Equation 3 is non-negative. That is, we require

$$\frac{\phi_0}{\phi_1} \geq \frac{\sigma_0}{\sigma_1} \exp\left[-\frac{1}{2}\frac{(\mu_0 - \mu_1)^2}{\sigma_1^2 - \sigma_0^2}\right]. \tag{6}$$

This assumption simply guarantees the existence of intersection points of the two scaled density functions; provided this holds, we establish an additional condition on priors necessary for $t^*(\phi_0/\phi_1)$ to be the average risk minimizer.

**Theorem B.1.** *If Equation 6 holds, define $t^*(\phi_0/\phi_1)$ as in Equation 3. Then, $t^*(\phi_0/\phi_1)$ is the statistical risk minimizer for the Gaussian mixture model if*

$$\frac{\phi_0}{\phi_1} > \Phi\left(\frac{t^*(\phi_0/\phi_1) - \mu_1}{\sigma_1}\right) \bigg/ \Phi\left(\frac{t^*(\phi_0/\phi_1) - \mu_0}{\sigma_0}\right). \tag{7}$$

*Proof.* For a decision rule $t \in \mathbb{R} \cup \{\pm\infty\}$, the statistical risk in the given Gaussian mixture model is given in Equation 1. The global minimum is achieved either at $\pm\infty$ or on $\mathbb{R}$. In the latter case, the minimizer is a solution to $\partial R(t)/\partial t = 0$:

$$0 = \frac{\partial R(t)}{\partial t} = -\frac{\phi_0}{\sqrt{2\pi}\sigma_0}\exp\left[-\frac{1}{2}\left(\frac{\mu_0 - t}{\sigma_0}\right)^2\right] + \frac{\phi_1}{\sqrt{2\pi}\sigma_1}\exp\left[-\frac{1}{2}\left(\frac{t - \mu_1}{\sigma_1}\right)^2\right].$$

Rearranging and taking the logarithm on both sides yields

$$0 = -2\log\left[\frac{\phi_0\sigma_1}{\phi_1\sigma_0}\right] - \left(\frac{t - \mu_1}{\sigma_1}\right)^2 + \left(\frac{\mu_0 - t}{\sigma_0}\right)^2, \tag{8}$$

which is a quadratic equation in $t$ with solutions

$$t_\pm = \frac{\mu_0\sigma_1^2 - \mu_1\sigma_0^2 \pm \sigma_0\sigma_1\sqrt{(\mu_0 - \mu_1)^2 + 2(\sigma_1^2 - \sigma_0^2)\log\frac{\phi_0\sigma_1}{\phi_1\sigma_0}}}{\sigma_1^2 - \sigma_0^2}. \tag{9}$$

By repeating the same steps for an inequality rather than equality, we conclude that $0 < \partial R(t)/\partial t$ if and only if

$$0 < -2\log\left[\frac{\phi_0\sigma_1}{\phi_1\sigma_0}\right] - \left(\frac{t - \mu_1}{\sigma_1}\right)^2 + \left(\frac{\mu_0 - t}{\sigma_0}\right)^2 \tag{10}$$

similarly to Equation 8. This identity holds because the logarithm is a monotonically increasing function, preserving the inequality. Further expanding the right-hand side of Equation 10 and collecting similar terms, we arrive at a quadratic equation in $t$ with the leading (quadratic) coefficient $\sigma_0^{-2} - \sigma_1^{-2} > 0$. Hence, the right-hand side defines an upward-branching parabola with zeros given in Equation 9 when they exist (we assume they do owing to assumption in Equation 6). The derivative of the statistical risk is positive whenever the right-hand side of Equation 10 is, i.e., on intervals $(-\infty, t_-)$ and $(t_+, \infty)$. Hence, the risk $R(t)$ must be increasing on the interval $(-\infty, t_-)$, and so $t_-$ can never be a global minimizer. Likewise, the risk is increasing on the interval $(t_+, \infty)$, which rules out $\{+\infty\}$. Therefore, we just need to establish that $R(-\infty) \geq R(t_+)$, which is equivalent to Equation 7 since $t^*(\phi_0/\phi_1) = t_+$. $\square$

**Remark.** We have considered unequal variances $\sigma_0^2 < \sigma_1^2$ as a natural way to model classes with different difficulty. Yet note that our analysis still holds with slight modifications when $\sigma_0 = \sigma_1 = \sigma$.

The difference in this case is that for any choice of priors, there is exactly one solution to Equation 8 (and exactly one intersection point of the scaled density functions), given by

$$t^*(\phi_0/\phi_1) = \frac{2\sigma^2 \log\left[\frac{\phi_0}{\phi_1}\right] + (\mu_1^2 - \mu_0^2)}{2(\mu_1 - \mu_0)}.$$

In particular, no additional assumptions as in Equation 6 to guarantee the existence of an intersection point need to be made. Furthermore, a similar derivative analysis as above implies that the risk is decreasing on the interval $(-\infty, t^*(\phi_0/\phi_1))$ and increasing on the interval $(t^*(\phi_0/\phi_1), +\infty)$, so that $t^*(\phi_0/\phi_1)$ must be the statistical risk minimizer and the assumption in Equation 7 is, in fact, always satisfied. With these simplifications, the rest of the analysis presented in this section and in Section 4 holds for $\sigma_0 = \sigma_1$.

## B.2 Worst-class optimal priors

We wish to formally establish that $t^*(\sigma_0/\sigma_1)$ as defined in Equation 4 minimizes both the average and worst-class risks when $\phi_0 \propto \sigma_0$ and $\phi_1 \propto \sigma_1$. For the first part, as we argued in Section B.1, these priors must satisfy the assumption in Equation 7 (Equation 6 is trivially satisfied). In this case, we can equivalently rewrite it as

$$\frac{\sigma_0}{\sigma_1} > \Phi\left(\frac{\mu_0\sigma_1 + \mu_1\sigma_0 - \mu_1\sigma_0 - \mu_1\sigma_1}{\sigma_1(\sigma_0 + \sigma_1)}\right) \Big/ \Phi\left(\frac{\mu_0\sigma_1 + \mu_1\sigma_0 - \mu_0\sigma_0 - \mu_0\sigma_1}{\sigma_1(\sigma_0 + \sigma_1)}\right)$$

$$= \Phi\left(\frac{\mu_0 - \mu_1}{\sigma_0 + \sigma_1}\right) \Big/ \Phi\left(\frac{\mu_1 - \mu_0}{\sigma_0 + \sigma_1}\right) = \left[1 - \Phi\left(\frac{\mu_1 - \mu_0}{\sigma_0 + \sigma_1}\right)\right] \Big/ \Phi\left(\frac{\mu_1 - \mu_0}{\sigma_0 + \sigma_1}\right),$$

Defining $z = \Phi\left(\frac{\mu_1 - \mu_0}{\sigma_0 + \sigma_1}\right)$, we arrive at $\sigma_0 z > \sigma_1(1 - z)$. By rearranging and collecting similar terms, we get $z > \frac{\sigma_1}{\sigma_0 + \sigma_1}$, which is equivalent to

$$\mu_1 - \mu_0 > (\sigma_0 + \sigma_1)\Phi^{-1}\left(\frac{\sigma_1}{\sigma_0 + \sigma_1}\right) \tag{11}$$

since $\Phi^{-1}$ is monotonically increasing. Hence, the assumption in Equation 7 can be interpreted as a lower bound on the separation between the two means $\mu_0$ and $\mu_1$. When this condition holds, $t^*(\sigma_0/\sigma_1)$ minimizes the average statistical risk, as desired.

For the second part, we start by proving the following lemma.

**Lemma B.2.** *Suppose $f : \mathbb{R} \to [0, 1]$ is a strictly increasing continuous function and $g : \mathbb{R} \to [0, 1]$ is a strictly decreasing continuous function, satisfying*

$$\begin{cases} \lim_{x \to -\infty} f(x) = \lim_{x \to +\infty} g(x) = 0 \\ \lim_{x \to +\infty} f(x) = \lim_{x \to -\infty} g(x) = 1 \end{cases} \tag{12}$$

*The solution $x^*$ to $\min_x \max\{f(x), g(x)\}$ is unique and satisfies $f(x^*) = g(x^*)$.*

*Proof.* Define $h(x) = f(x) - g(x)$. Observe that $h(x)$ is a strictly increasing continuous function as $f$ and $-g$ are strictly increasing. Also, $\lim_{x \to -\infty} h(x) = -1$ and $\lim_{x \to \infty} h(x) = 1$. From the Intermediate Value Theorem, there exists a point $x^*$, where $h(x^*) = 0$, i.e., $f(x^*) = g(x^*)$. Observe that $x < x^*$ implies $\max_x\{f(x), g(x)\} = g(x)$ and $x \geq x^*$ implies $\max_x\{f(x), g(x)\} = f(x)$. Therefore, the objective function decreases up to $x^*$ and then increases. Hence, $x^*$ is the minimizer of the objective $\min_x \max\{f(x), g(x)\}$. $\square$

Since $f(t) = R_1(t)$ and $g(t) = R_0(t)$ meet the conditions of Lemma B.2, the minimizer of the worst-class risk $\hat{t}$ satisfies $R_0(\hat{t}) = R_1(\hat{t})$. Equating the risks in Equation 1 then immediately proves the formula for $\hat{t}$ (Equation 4). Finally, note that $t^*(\sigma_0/\sigma_1) = \hat{t}$ because of the vanishing logarithm. Therefore, $t^*(\sigma_0/\sigma_1)$ minimizes both the average and worst-class statistical risks for a mixture with priors $\phi_0 = \frac{\sigma_0}{\sigma_0 + \sigma_1}$ and $\phi_1 = \frac{\sigma_1}{\sigma_0 + \sigma_1}$ provided that Equation 11 (reformulated assumption in Equation 7 for the given choice of priors) holds.

Finally, note that if $\sigma_0 = \sigma_1$, $\hat{t} = t^*(\phi_0/\phi_1) = (\mu_0 + \mu_1)/2$ for the optimal priors $\phi_0 = \phi_1$, as expected.

## B.3 THE EFFECT OF SSP

Our setting allows us to adapt and analyze a state-of-the-art baseline pruning algorithm, *Self-Supervised Pruning (SSP)* (Sorscher et al., 2022). Recall that, in Section 4, we adopted a variant of SSP that removes samples within a margin $M > 0$ of their class means. SSP performs k-means clustering in the embedding space of an ImageNet pre-trained self-supervised model and defines the difficulty of each data point by the cosine distance to its nearest cluster centroid, or prototype. In the case of two univariate Gaussians, this score corresponds to measuring the distance to the closest mean.

We claimed and demonstrated in Figure 3 that the optimal risk minimizer remains around $t^*(\phi_0/\phi_1)$ even after pruning. In this section, we make these claims more precise. To this end, note that, for a Gaussian variable with variance $\sigma^2$, the removed probability mass is

$$\Phi\left(\frac{M + \mu - \mu}{\sigma}\right) - \Phi\left(\frac{\mu - (\mu - M)}{\sigma}\right) = 2\Phi\left(\frac{M}{\sigma}\right) - 1. \tag{13}$$

Now, for sufficiently small $M$, we can assume that the average risk minimizer lies within the interval $(\mu_0 + M, \mu_1 - M)$ (recall that the average risk minimizer of the original mixture lies between $\mu_0$ and $\mu_1$ owing to the assumption in Equation 7). In this case, the right tail of the easier class and the left tail of the more difficult class (these tails are misclassified for the two classes) are unaffected by pruning and, hence, the average risk after SSP $R'(t)$ should remain proportional to the original risk $R(t)$. More formally, consider the class-wise risks after SSP:

$$R'_0(t) = \frac{\Phi\left(\frac{\mu_0 - t}{\sigma_0}\right)}{2 - 2\Phi\left(M/\sigma_0\right)} = \frac{R_0(t)}{2 - 2\Phi\left(M/\sigma_0\right)},$$

$$R'_1(t) = \frac{\Phi\left(\frac{t - \mu_1}{\sigma_1}\right)}{2 - 2\Phi\left(M/\sigma_1\right)} = \frac{R_1(t)}{2 - 2\Phi\left(M/\sigma_1\right)}.$$

where the denominators are the normalizing factors based on Equation 13. To compute the average risk $R'(t) = \phi'_0 R'_0(t) + \phi'_1 R'_1(t)$, we shall identify the modified class priors $\phi'_0$ and $\phi'_1$ after pruning. Again, from Equation 13, we obtain

$$\phi'_0 \equiv \phi_0\left[2 - 2\Phi\left(\frac{M}{\sigma_0}\right)\right], \qquad \phi'_1 \equiv \phi_1\left[2 - 2\Phi\left(\frac{M}{\sigma_1}\right)\right] \tag{14}$$

up to a global normalizing constant that ensures $\phi'_0 + \phi'_1 = 1$. Therefore, the average risk is indeed proportional to $\phi_0 R_0(t) + \phi_1 R_1(t) = R(t)$, as desired, so the average risk minimizer after pruning coincides with the original one.

## B.4 MULTIVARIATE ISOTROPIC GAUSSIANS

While the analysis of general multivariate Gaussians is beyond the scope of this paper, we show here that in the case of two *isotropic* multivariate Gaussians (i.e., $\text{Var}(x|y) = \sigma_y^2 I$ for $y = 0, 1$), the arguments in Sections B.1–B.3 apply. In particular, we demonstrate that this scenario reduces to a univariate case.

For simplicity, assume that the means of the two Gaussians are located at $-\mu$ and $\mu$ for some $\mu \in \mathbb{R}^d$ for classes $y = 0$ and $y = 1$, respectively, which can always be achieved by a distance-preserving transformation (translation and/or rotation). Generalizing the linear classifier from Section 4, we now consider a hyperplane defined by a vector $w \in \mathbb{R}^d$ of unit $\ell_2$-norm and a threshold $t \in \mathbb{R} \cup \{\pm\infty\}$, classifying a point $x \in \mathbb{R}^d$ as $\mathbf{1}\{w^\top x \leq t\}$. Note that for a fixed vector $w \in \mathbb{R}^d$ and $x \sim \mathcal{N}(\mu, \sigma^2 I_d)$, we have $(w^\top x - w^\top \mu)/\sigma \sim \mathcal{N}(0, 1)$ as a linear transformation of the isotropic Gaussian. We can now compute the class risks as

$$R_1(w, t) = P(w^\top x \leq t | y = 1)$$
$$= P\left(\frac{w^\top x - w^\top \mu}{\sigma_1} \leq \frac{t - w^\top \mu}{\sigma_1} \middle| y = 1\right) = \Phi\left(\frac{t - w^\top \mu}{\sigma_1}\right),$$
$$R_0(w, t) = P(w^\top x \geq t | y = 0)$$
$$= P\left(\frac{w^\top x + w^\top \mu}{\sigma_0} \geq \frac{t + w^\top \mu}{\sigma_0} \middle| y = 1\right) = \Phi\left(\frac{-t - w^\top \mu}{\sigma_0}\right).$$

The average risk given the class priors is thus

$$R(w,t) = \phi_1 \Phi\left(\frac{t - w^\top \mu}{\sigma_1}\right) + \phi_0 \Phi\left(\frac{-t - w^\top \mu}{\sigma_0}\right).$$

We will now show that for all the risk minimization problems we have considered in our theoretical analysis, we can equivalently minimize over a one-dimensional problem. It suffices to observe that for a fixed $t \in \mathbb{R}$, both $R_0(w,t)$ and $R_1(w,t)$ are minimized when $w$ (with $\|w\|_2 = 1$) coincides with $\mu/\|\mu\|$ because of monotonicity of $\Phi$. This leads to the expressions in Equation 1 with $\mu_{1,0} = \pm\mu/\|\mu\|$. Thus, to compute the minimum of $R(w,t)$ with respect to $w$ and $t$, we can equivalently minimize $R(\mu/\|\mu\|_2, t)$ with respect to $t$. For the worst-class optimization

$$\min_{\|w\|=1,t} \max\{R_1(w,t), R_0(w,t)\}$$

observe that

$$\min_t \min_{\|w\|=1} \max\{R_1(w,t), R_0(w,t)\} = \min_t \max\{R_1(\mu/\|\mu\|, t), R_0(\mu/\|\mu\|, t)\}$$

Therefore, this problem also reduces to the minimization in the univariate case from Section B.2.

## C  DATA PRUNING: FULL RESULTS

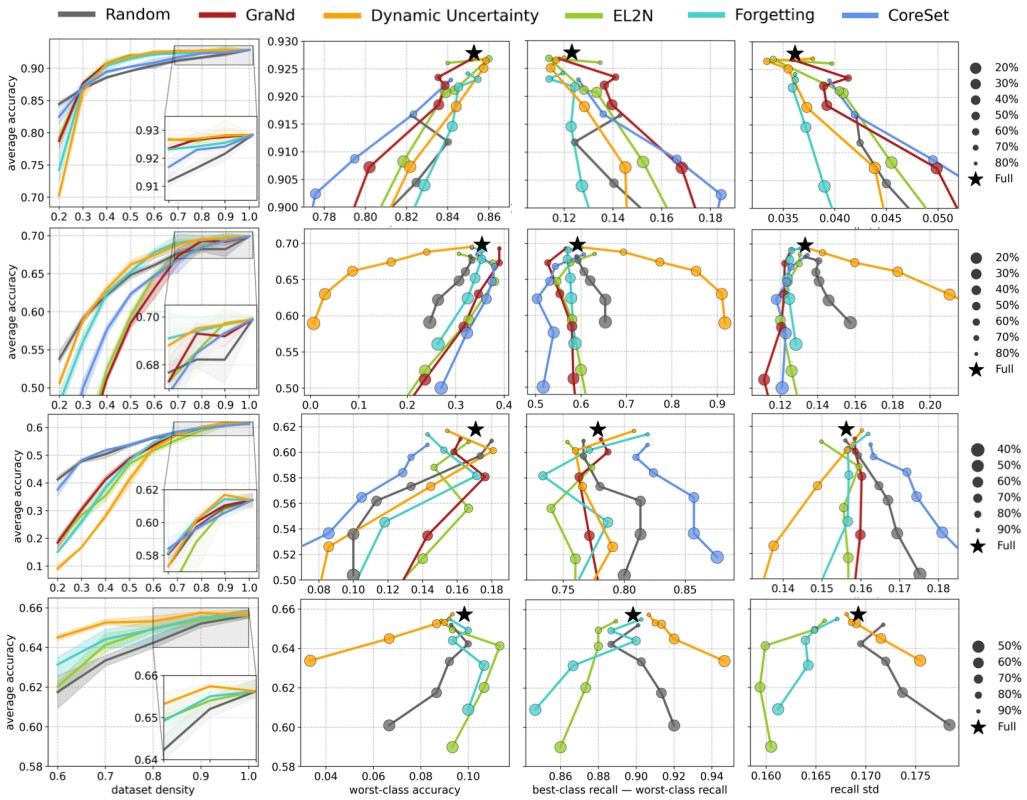

Figure 10: The average test performance of various data pruning algorithms against dataset density (fraction of samples remaining after pruning) and metrics of class robustness. **Top to Bottom**: VGG-16 on CIFAR-10; VGG-19 on CIFAR-100; ResNet-18 on TinyImageNet; ResNet-50 on ImageNet. All results averaged over 3 random seeds. Error bands represent min/max.

# D DRoP: FULL RESULTS

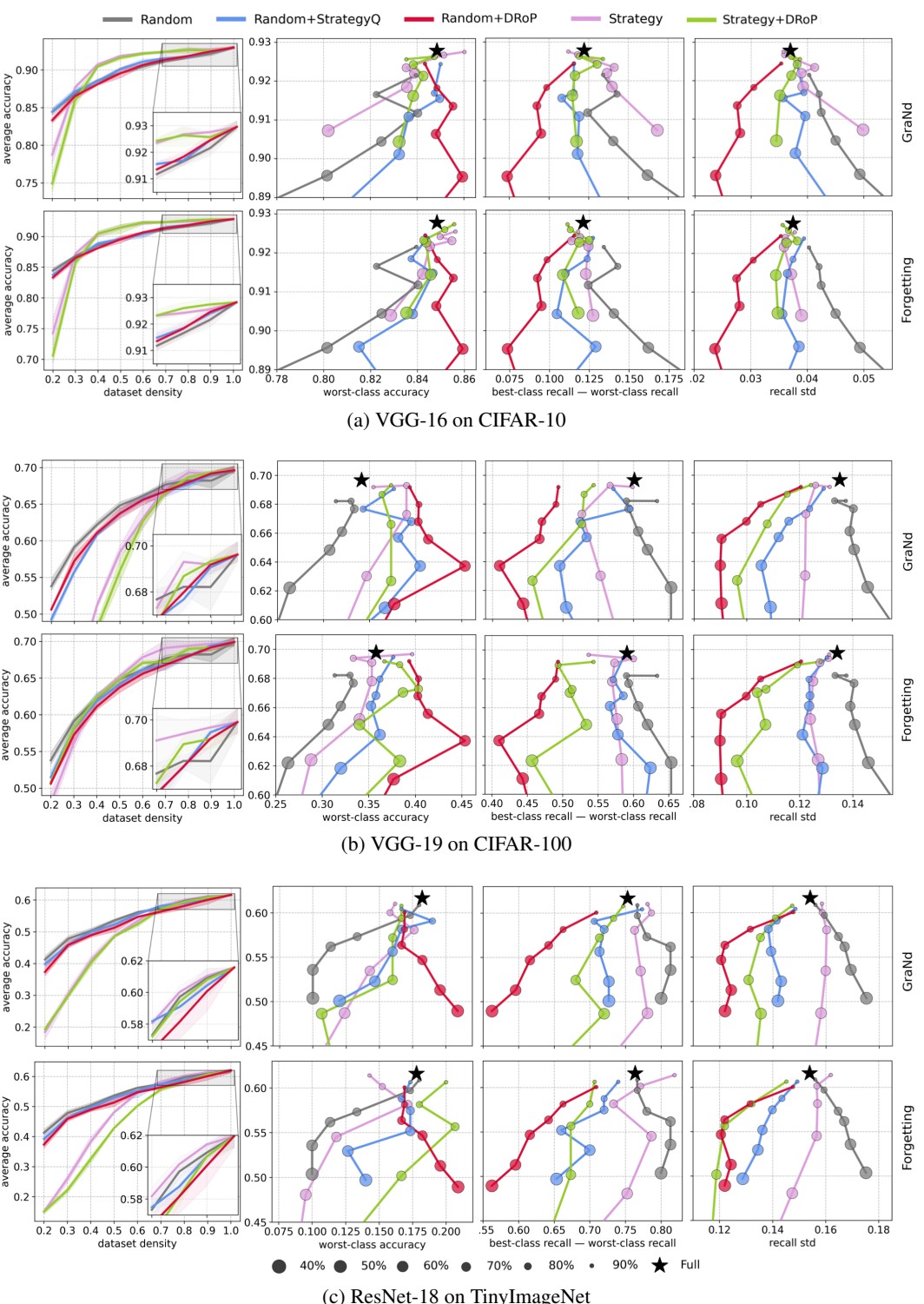

Figure 11: The average test performance of various data pruning protocols against dataset density and measures of class robustness. Random+DroP consistently outperforms all baselines with respect to these additional measures of distributional robustness besides worst-class accuracy reported in Figure 5, too, confirming its effectiveness in reducing bias. All results averaged over 3 random seeds. Error bands represent min/max.

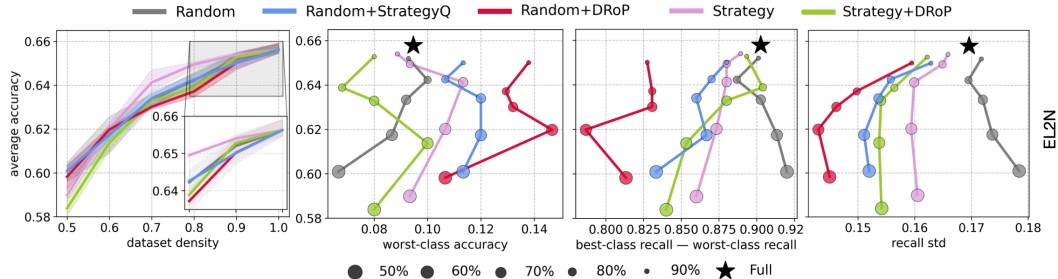

Figure 12: The average test performance of various data pruning protocols against dataset density and measures of class robustness. ResNet-50 on ImageNet with EL2N as Strategy. All results averaged over 3 random seeds. Error bands represent min/max.

Table 2: Average and worst-class accuracy (in %) when pruning datasets to $d = 0.3, 0.5, 0.8$ by various methods with or without DRoP. For each column, we boldface the best worst-class accuracy across all three methods. In all cases, DRoP achieves superior worst-class accuracy, especially so when used together with Random. Furthermore, Random+DRoP always improves robustness of full datasets when halving them ($d = 0.5$).

| Method | DRoP | Acc. | CIFAR-10 | | | CIFAR-100 | | | TinyImageNet | | |
|---|---|---|---|---|---|---|---|---|---|---|---|
| | | | 0.3 | 0.5 | 0.8 | 0.3 | 0.5 | 0.8 | 0.3 | 0.5 | 0.8 |
| Random | ✗ | avg | 86.8 | 89.6 | 91.7 | 59.1 | 64.8 | 68.2 | 47.8 | 53.6 | 59.7 |
| | | worst | 75.5 | 80.1 | 82.2 | 24.7 | 30.1 | 33.0 | 8.7 | 10.0 | 17.3 |
| | ✓ | avg | 86.5 | 89.5 | 91.8 | 57.2 | 63.7 | 68.0 | 45.8 | 51.3 | 58.1 |
| | | worst | **82.1** | **85.9** | 84.9 | **34.3** | **45.3** | **40.3** | **16.2** | **20.0** | 16.9 |
| GraNd | ✗ | avg | 87.6 | 91.9 | 92.7 | 40.4 | 58.5 | 69.3 | 30.4 | 48.7 | 60.0 |
| | | worst | 77.0 | 83.6 | 85.1 | 12.3 | 31.7 | 39.0 | 1.0 | 12.4 | 15.8 |
| | ✓ | avg | 85.9 | 91.6 | 92.6 | 35.5 | 55.9 | 68.7 | 29.9 | 48.6 | 59.4 |
| | | worst | 78.0 | 83.8 | 84.7 | 15.0 | 30.1 | 36.3 | 4.7 | 10.7 | 16.7 |
| Forgetting | ✗ | avg | 87.2 | 91.5 | 92.4 | 56.1 | 65.2 | 69.4 | 25.9 | 48.1 | 60.2 |
| | | worst | 79.1 | 84.2 | 85.0 | 26.3 | 34.0 | 33.3 | 1.0 | 9.5 | 15.2 |
| | ✓ | avg | 85.7 | 91.4 | 92.6 | 58.2 | 64.8 | 68.9 | 22.1 | 42.8 | 58.2 |
| | | worst | 78.7 | 84.6 | **85.1** | 32.0 | 34.0 | 38.3 | 1.3 | 12.7 | **18.0** |
| Average | ✗ | avg | 87.2 | 91.0 | 92.3 | 51.9 | 62.8 | 69.0 | 34.7 | 50.1 | 60.0 |
| | | worst | 77.2 | 82.6 | 84.1 | 21.1 | 31.9 | 35.1 | 3.6 | 10.6 | 16.1 |
| | ✓ | avg | 86.0 | 90.8 | 92.3 | 50.3 | 61.5 | 68.5 | 32.6 | 47.6 | 58.6 |
| | | worst | 79.6 | 84.8 | 84.9 | 27.1 | 36.5 | 38.3 | 7.4 | 14.5 | 17.2 |
| Full Dataset | | avg | | 92.8 | | | 70.0 | | | 61.6 | |
| | | worst | | 84.8 | | | 35.8 | | | 17.8 | |

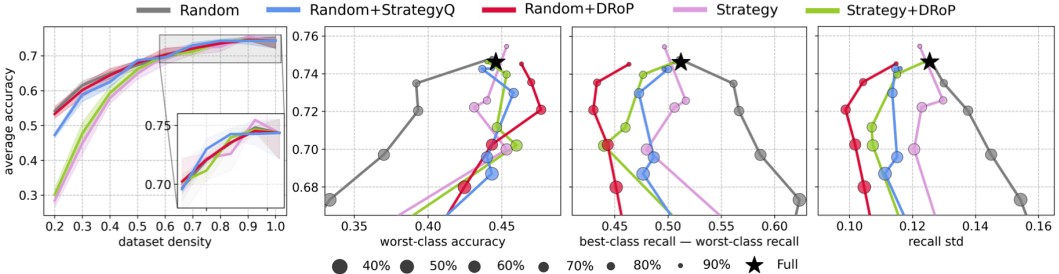

Figure 13: The average test performance of various data pruning protocols against dataset density and measures of class robustness. Wide-ResNet-101 on CIFAR-100 with EL2N as Strategy. All results averaged over 3 random seeds. Error bands represent min/max.

**Larger Models.** Some previous studies argue that larger models enjoy better distributional robustness compared to smaller ones (Andreassen et al., 2022; Feuer et al., 2023). To see how model size could affect the applicability and effectiveness of DRoP, we train a Wide-ResNet-101 (Zagoruyko, 2016) with expansion factor of 2 on CIFAR-100, which is an unusually high caliber architecture for such a relatively small dataset. In fact, we had to remove a few early downsampling layers from the original implementation of this model to account for the resolution of CIFAR-100. On full dataset, Wide-ResNet-101 with 125M parameters achieves a $4\%$ higher average and almost a $10\%$ higher worst-class accuracy compared to a VGG-19 with just 20M parameters (see Figures 11b and 13). Even in this more competitive setup, Random+DRoP continues to show considerable improvement in all robustness metrics over other pruning baselines as well as the full dataset (Figure 13).

## E  DRoP ABLATION: PRE-TRAINING LENGTH

As mentioned in Section 5, DRoP requires a pre-trained query model to compute class-wise validation errors. All existing data pruning methods also require pre-training to compute sample-wise pruning scores. However, as Paul et al. (2021) shows, it suffices to run optimization for as little as 10 epochs to achieve accurate enough scores for EL2N. In our experimental work, we used $10\%$ of the full training cycle to train the query model for EL2N, GraNd, and DRoP, which is at least the recommended 10 epochs for all model-dataset pairs tested in this study. In this section, we verify that DRoP reaches its full performance with this much pre-training and, hence, it does not introduce any additional computational overhead already needed by the current data pruning algorithms. Indeed, Figure 14 shows that the downstream performance of DRoP saturates at 10 pre-training epochs or earlier, with lower pruning ratios allowing even less training than more aggressive ones.

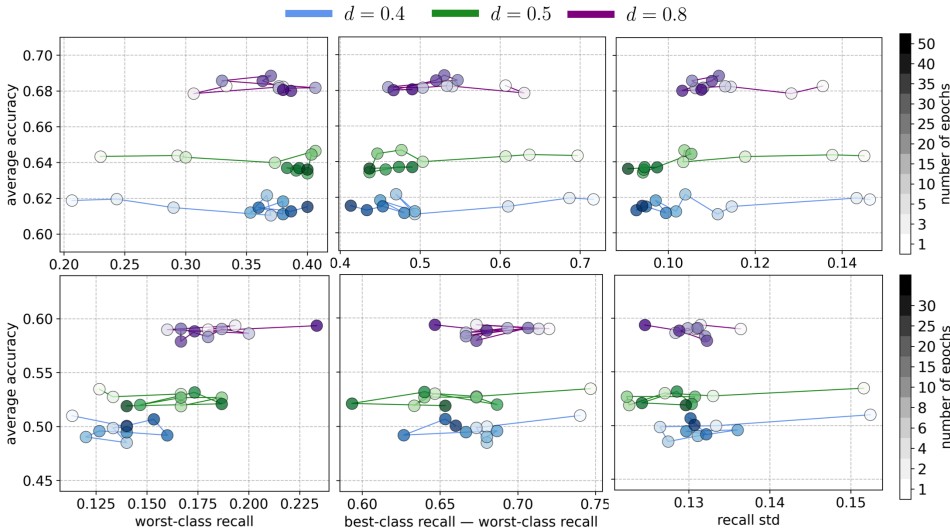

Figure 14: Final model performance after DRoP with different lengths of the query model training. **Top:** VGG-19 on CIFAR-100, **Bottom:** ResNet-18 on TinyImageNet. Repeated over 3 random seeds and three dataset densities ($d = 0.4, 0.5, 0.8$). In our study, we use 16 and 9 pre-training epochs for VGG-19 on CIFAR-100 and ResNet-18 on TinyImageNet ($10\%$ of the full training cycle), respectively.

## F  RESNET-50 ON WATERBIRDS

As stated in the caption of Figure 9, Sagawa* et al. (2020) compute average accuracy on Waterbirds as sum of group accuracy weighted by the group proportions in the training dataset, which is highly skewed towards majority classes (landbird+forest and waterbird+water, see Figure 15b). We adhere to this strategy in Figure 9 to ensure fair comparison with prior algorithms. Note, however, that this places DRoP at a disadvantage since it is designed to prune easier groups more aggressively and protect underperforming minority groups (see left barplots for each method in Figure 15b). Ultimately,

this leads to a rapid degradation of the average accuracy at low dataset densities, which we observed in Figure 9 (left). In Figure 15a (bottom) we switch to a more standard accuracy computation as fraction of correctly classified samples regardless of group identities. In this framework, Random+DRoP remains strong relative to other methods. We should note that the test split is also slightly imbalanced with landbird+forest and landbird+water represented by 2255 images each, and waterbird+forest and waterbird+water—by 642.

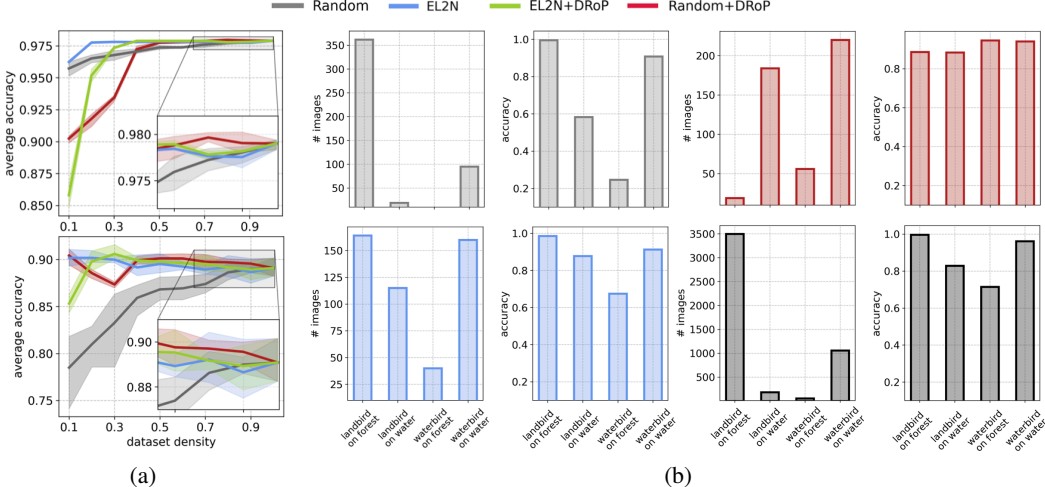

Figure 15: **(a)**: Average test accuracy against dataset density across different pruning methods. **Top:** Average accuracy computed as a weighted sum of group-wise accuracies according to training proportions. **Bottom:** Average accuracy is computed as a fraction of correctly classified samples of the test set. **(b)**: Group sizes and group accuracy for all four groups in Waterbirds at $d = 0.1$ for different pruning methods, as well as on the full dataset (bottom right barplots).

# G   DATA PRUNING: CLASS-WISE ANALYSIS

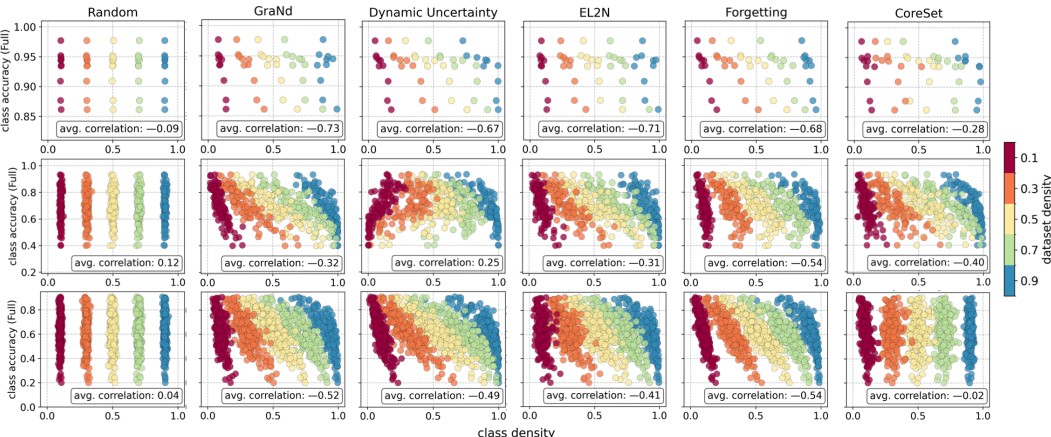

Figure 16: Full dataset test accuracy against density, across all classes, after pruning with different methods. We use the full dataset accuracy to capture the baseline "difficulty" of each class. On each plot, we report the correlation coefficient between these two quantities across classes, averaged over 5 data density levels (0.1, 0.3, 0.5, 0.7, 0.9). **Top:** VGG-16 on CIFAR-10, **Center:** VGG-19 on CIFAR-100, **Bottom:** ResNet-18 on TinyImageNet. Experiments repeated over 3 random seeds.

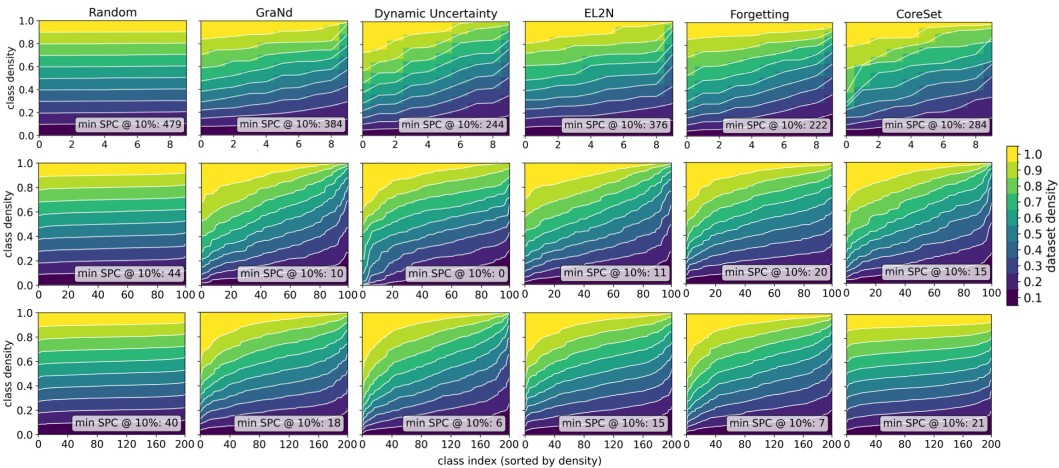

Figure 17: Sorted class densities in the training dataset pruned by various algorithms to different density levels. On each plot, we report the minimum number of samples per class (SPC) at 10% dataset density. **Top:** VGG-16 on CIFAR-10, **Center:** VGG-19 on CIFAR-100, **Bottom:** ResNet-18 on TinyImageNet. Repeated over 3 random seeds.

