# OpenReview forum: "DRoP: Distributionally Robust Data Pruning"
_ICLR.cc/2025/Conference — ICLR 2025 Spotlight_

### Official Review · Reviewer_kAs6 · 2024-11-03

**Soundness:** 3
**Presentation:** 3
**Contribution:** 3
**Rating:** 8
**Confidence:** 2

**Summary:**

The paper studies classification bias in data pruning. It first provides a systematic study of the existing methods and shows that they are all biased to some extent. To this end, the authors propose DRoP, a distributionally robust approach for pruning that preserves good robustness against classification bias. Their method achieves state-of-the-art robustness performance when combined with random subsampling across different settings.

**Strengths:**

- The proposed method DRoP is quite concise, which is highly appreciated since it can potentially increase the accessibility and applicability of the method beyond the current scope.
- Despite its simplicity, DRoP + random sampling achieves superior performance across various settings.
- The paper conducts a systematic evaluation of the existing data pruning methods and reveals a valuable conclusion that none of the existing methods actually has good robustness in terms of classification bias.
- The evaluation covers a wide range of scenarios beyond the classic data pruning, which enhances the importance of the method as it could lead to potential influence that extends beyond traditional use cases.
- The discussion of the limitations and future work is sound and makes sense.

**Weaknesses:**

- The flow of the paper can be improved. At the moment it feels that the related work is scattered throughout the paper, which disrupts readability and makes it challenging to grasp. It would be clearer to bring all of the related work into one section and then refer back to it when needed in the experiment section. For example, the authors may group related work into categories like "Data Pruning Methods", "Robustness and Fairness", "Long-Tailed Recognition Techniques", and "Robust Data Pruning Methods", and place this consolidated section after the introduction.

- Since there is no standalone related work section, it is difficult to assess the novelty of the proposed method and the relation with existing data pruning techniques. It would be great to also include a brief discussion of the most recent advancements in data pruning and how DRoP is similar to/different from them.

- The robustness evaluation on imbalanced datasets and group distributional robustness is not compared with the methods from the corresponding fields. It would be great if the performance of some task-specific methods is also compared in Figures 8 and 9, which helps to better assess the method's relative performance in context. For example, [1,2] for "imbalanced datasets" and [3,4] for "group distributional robustness"

[1] Kang, Bingyi, et al. "Decoupling representation and classifier for long-tailed recognition." ICLR 2020.

[2] Cao, Kaidi, et al. "Learning imbalanced datasets with label-distribution-aware margin loss." NeurIPs 2019.

[3] Liu, Evan Z., et al. "Just train twice: Improving group robustness without training group information." ICML 2021.

[4] Sagawa, Shiori, et al. "Distributionally robust neural networks for group shifts: On the importance of regularization for worst-case generalization." ICLR 2020.

**Questions:**

- It is quite impressive that Random + DRoP outperforms other methods when the overall dataset density is low (e.g. in Figure 5). But do the authors have an idea why in many cases, it gives inferior worst-class performance when the overall dataset density is high? For example, for ResNet18 + TinyImageNet at 90% density, Random + DRoP is the second worst performance (only better than Forgetting) in terms of the worst-class performance in Figure 5.

---

> ### Author Response · Authors · 2024-11-15
> **Authors' Response**
>
> We thank the reviewer for your strong eveluation of our work and thoughtful comments. We respond to them below.
>
> 1. We agree with your suggestion, and we revised th manuscript to include a separate "Related Work" section (see point 2 of the General Response).
>
> 2. We hope that the discussion of prior art in data pruning and distributional robustness is now easier to identify with a standalone "Related Works" section that we added.
>
> 3. We agree that we include only limited cross-evaluation with existing methods in distributional robustness and learning on long-tailed datasets primarily because we attribute our contributions to data pruning; we state this explicitly in the Limitations paragraph on page 10. However, we added many more baseline methods to Figure 9 (Waterbirds)--please see point 4 of the General Response as well as the revised manuscript.
>
> We thank you once again for your encouraging feedback, and we hope that you will find all your suggestions well addressed in the revision.

---

> > ### Comment · Reviewer_kAs6 · 2024-12-01
> >
> > Thank you for the rebuttal and efforts to add more baselines. Now the paper is even clearer and reads more smoothly after the reorganization. Good luck with your submission!

---

### Official Review · Reviewer_XsKC · 2024-11-04

**Soundness:** 4
**Presentation:** 4
**Contribution:** 4
**Rating:** 6
**Confidence:** 4

**Summary:**

The paper studies the effect of dataset pruning on classification bias for several existing methods using the metrics of worst-class accuracy, difference between best and worst class accuracy, and standard deviation. Due to the purported worsening of worst-class accuracy at the cost of better average performance, the paper proposes random pruning based on error rates on a holdout set which is motivated by a theoretical analysis of a simple 1-D mixture of two Gaussians. Experimental analysis is conducted on CIFAR10, CIFAR100, TinyImageNet and ImageNet datasets for the various settings.

**Strengths:**

1. The presented theoretical analysis on the GMM for minimizing worst-case statistical risk and average risk is good and aligns with the proposed pruning method.
2. The experimental analysis is comprehensive and convincing.

**Weaknesses:**

1. While the Random+DRoP method does reduce classification bias compared to existing dataset pruning strategies, it generally seems to do slightly worse on average accuracy (Figure 5). The combined Strategy+DRoP seems to mitigate this a bit but the trend is not clear.
2. The experiments do not always indicate such a clear and striking trend towards robustness as the authors suggest. Some aggregate scalar numbers might be beneficial to understand which strategy does better across all data densities.

**Questions:**

1. in Figure 6, right, why does a higher negative correlation of class accuracy to class density indicate more robustness? Ideally class accuracy should have zero correlation with class density for a model invariant to class size.

---

> ### Author Response · Authors · 2024-11-15
> **Authors' Response (1/2)**
>
> We thank the reviewer for a thorough and encouranging assessment of our work. We address your comments below in two parts.
>
> 1. We would argue that the average and worst-class/group performance are in natural clash with each other, and it is accepted that improvements in the former almost inevitably lead to degradation of the latter (Chaudhuri et al., 2023). Prior art in DRO directly optimizes for worst-group accuracy alone, willing to accept lower average accuracy in return for improved robustness (Sagawa et al., 2020). Our work provides an intuitive explanation for this trade-off: the most difficult classes of the data distribution need more resources (in our case--larger presence in the dataset) to be well-learned, requiring us to shift our focus from and sacrifice some performance of the easier ones. While this reasoning might seem overly simplistic by portraying individual classes as antagonists playing a zero-sum game for resources, at the same time, it is realistic that different classes require different features to be learned by a fixed-capacity model and hence compete with one another (Aguilar-Ruiz, 2024). Our theoretical analysis provides a unique illustration of the trade-off between the average and worst-class performance, increasing the value of our contributions to the community. Further, our plots are designed to best visualize this trade-off across different methods by plotting results in the coordinates of both average and worst-class accuracy. We believe that they consistently show that DRoP traces a better trade-off than other methods, especially when used together with random pruning (Random+DRoP).
>
>     Thus, average accuracy against datset density (leftmost plots in Figure 5) is not our key performance indicator, but we agree that it must be studied as well. Your observation that EL2N+DRoP mitigates some of the average accuracy degradation incurred by Random+DroP is related to the differences between score-based (e.g., EL2N) and random data pruning and depends on the compression rates. Sorcher et al. (2022) found that it is beneficial to retain more difficult examples picked by score-based methods only when pruning is not aggressive, whereas random pruning or inverse score-based pruning that retains easier samples is better at higher rates. Our results with and without DRoP confirm this phenomenon: pruning backed by score-based methods improves over random (Random, Random+DRoP, Random+StrategyQ) for high densities but quickly deteriorates for lower densities. Thus, your observation is valid in the high-density regime but not in low-density regime, which does indeed obscure the trends. In any case, we argue that this phenomenon is not related to DRoP but is idiosyncratic to existing data pruning  methods; note that Random and Random+DRoP are almost indistinuishable on the leftmost plots in Figure 5, as well as EL2N and EL2N+DRoP.

---

> ### Author Response · Authors · 2024-11-15
> **Authors' Response (2/2)**
>
> 2. We added a table reporting mean values for the majority of our experiemnts (see Table 2 in Appendix D). For each dataset and density, DRoP achieves the best worst-class accuracy across all tested baselines.
>
> 3. For each class, Figure 6 (right) plots its validation recall achieved by the preliminary model fully trained on the entire dataset against its density in the DRoP-pruned dataset, for a variety of overall dataset densities that are color-encoded. Intuitively, the more difficult a class is as measured by the preliminary model (smaller ordinate), the higher its density should be in a robust pruned dataset (larger abscissa), which is a central idea behind DRoP. This "rule" is suggested by our theoretical analysis in Section 4 where we substitute a provable optimal ratio of class standard deviations for the ratio of class errors achieved by the average risk minimizer (see Figure 4 and the discussion in the last paragraph of Section 4). Thus, a more pronounced negative correlation in Figure 6 (right) conforms with this logic and is treated as desirable. Note that DRoP computes class densities to be exactly proportional to class errors as computed by the preliminary model, and the correlation is not identically -1 only because (1) DRoP does not execute a full training cycle of the preliminary model, producing noisy estimates of class errors (see Appendix E), and (2) some target class proportions computed by DRoP cannot be satisfied exactly because class density cannot exceed 1 (see lines 344-350 and note many classes with density capped at 1 in Figure 6).
>
> We hope that we fully answered your questions and addressed all your concerns with our responses and changes to the manuscript. We thank you once again for your positive evaluation and hope that you will consider increasing your rating of our paper.
>
> **References** \
> [1] Sagawa, S., et al. "Distributionally Robust Neural Networks". ICLR 2020. \
> [2] Aguilar-Ruiz, J. "Class-specific feature selection for classification explainability". 2024. \
> [3] Sorcher, B., et al. "Beyond Neural Scaling Laws: Beating Power Law Scaling via Data Pruning". NeurIPS 2022.  \
> [4] Chaudhuri, K., et al. "Why does Throwing Away Data Improve Worst-Group Error?". ICML 2023.

---

> > ### Author Response · Authors · 2024-11-26
> >
> > We have greatly appreciated the feedback from the reviewer and incorporated it into the paper. Since the rebuttal period is going to end soon, please let us know if you are happy with our responses and changes to the paper. We would be very grateful if you considered increasing the score.

---

### Official Review · Reviewer_JvuS · 2024-11-04

**Soundness:** 1
**Presentation:** 1
**Contribution:** 3
**Rating:** 8
**Confidence:** 3

**Summary:**

This paper presents a dataset pruning method that helps the models trained on the pruned datasets be more fair in their class accuracies. The proposed method is called "DRoP", and it selects class-specific pruning ratios based on the error rates, improving performance on underrepresented classes.
The emprical evaluations are performed using VGGs, ResNets on CIFAR-10, CIFAR-100, TinyImageNet, and WaterBirds datasets.
Most of the evaluations are very favourable for the proposed method.

**Strengths:**

The idea of DRoP is quite interesting and to the best of my understanding very novel.

The method seems to be simple and straightforward.

The paper provides detailed theoritical insights and this helped understand the method.

**Weaknesses:**

The framing of the paper is currently unclear to me.
(W1) Firstly, It would significantly help in include "Dataset Pruning" rather than just "Pruning" in the title of the paper, because given the dominence of "model pruning" methods, it is very understandable that one might get confused by the title. One suggestion would be DRoPD (.... Pruning Datasets).

(W2) Secondly, the problem of "classification bias" is never really defined in the paper, that is, it is never explained what is meant by "classification bias" and why is it a problem and how datasets affect this so-called "classification bias".

(W3) Next, the use of the word "Robustness" seems incorrect in this context. It is not really "robustness" being evaluated but class-wise fairness. Might be prudent to frame the paper differently around this.

(W4) Lastly, the evaluations are limited to non-robust small models like VGG-16, VGG-19, ResNets up to ResNet50. However, it has been seen that large models like ResNet101, ConvNeXt-B onwards, and ViT based large models are more robust to distribution changes. They might be more robust to underrepresentation of classes as well, or might not show gains when using the proposed "DRoP" method. Thus having experimental evaluations with these models would help better understand the applicability of the proposed method.

**Questions:**

Q1- In Figure 9 (leftmost plot), when dataset sparisty is more than 50% (density less than 0.5) why is the proposed DRoP significantly less stable than all the other methods? Some explaination for this would be very helpful.

---

> ### Author Response · Authors · 2024-11-15
> **Authors' Response (1/2)**
>
> We thank the reviewer for their time and feedback. We would like to address your concerns below (please note that we break our response into two parts due to space constraints)
>
> W1. While our abstract unequivocally specifies the type of pruning we are concerned with, we are happy to change the title to "DRoP: Distributionally Robust **Data** Pruning" to avoid any confusion, as per your suggestion.
>
> W2. We respectfully disagree that the problem of classification bias is not discussed and defined in our paper. In paragraph "Robustness and Evaluation Metrics" (Section 2, page 3), we first introduce and overview distributional robustness, and then define classification bias as a special case of that more general framework in lines 136-138. Further, in lines 138-143, we refer to existing literature to explain what methods and datasets are prone to classification bias (e.g., long-tailed datasets, model pruning, and adversarial training). To make this point clearer in the text, we made a few changes to the structure of our paper (see points 2 and 3 of the General Response).
>
> W3. It is true that we predominantly focus on the problem of classification bias, as we claim in Section 1, imply throughout the text, and explicitly state in lines 136-138 of Section 2 and in line 535 of Limitations. However, as we argue in Section 2 (lines 120-137), classification bias is a special case of *distributional* robustness (DRO). We state in lines 120-121 that "Distributional robustness in machine learning concerns the issue of non-uniform accuracy over the data distribution (Sagawa et al., 2020)", and, after a comprehensive discussion of topics and prior art of DRO, connect it to classification bias in lines 136-138: "The focus of our study is a special case of group robustness, classification bias, where groups are directly defined by class attributions". Our contributions are clearly equally applicable more generally across the domain of DRO, and we even evaluate DRoP on Waterbirds (see the last paragraph of Section 5)----a standard benchmark for group-wise robustness. Therefore, we argue that our paper and contributions are correctly framed around distributional robustness. Finally, we would like to point out that the suggested "class-wise fairness" is not an appropriate scope for our paper as "fairness" is a separate sub-field within DRO that focuses on demographic group attributes and optimizes for different metrics such as equal opportunity and equalized odds (see lines 126-128). We recognize, however, that the literature on DRO and related topics is extensive and interwoven, and we welcome your concrete suggestions on how we might better delineate these distinctions in our work.
>
> W4. We agree that expanding the set of architectures to even larger models like ViT could benefit the empirical section of the paper and strengthen our conclusions. Unfortunately, it is not feasible to conduct more experiments with these models on large-scale datasets they deserve in the timeframe of the rebuttal period. Thus, we are currently repeating some of our empirical work with ResNet-101 on CIFAR-100, and we will include these results should they be ready in time. Still, we would like to argue that the reported benchmarks cover a wide range of scales (CIFAR-10 to ImageNet), proving our empirics to be well on par or stronger compared to many reputed papers recently published in this field (Paul et al., 2021; Good et al., 2022; Ma et al., 2022; Ayed & Hayou, 2023; Zhao & Bilen, 2023; Feng et al., 2024). To further strenghten our empirical evidence, we incorporate imbalanced datasets (Figure 8), group-wise robustness benchmarks (Figure 9), as well as an alternative specialized optimization procedure from prior art (Figure 7). Overall, we believe that our empirical work is strong and consistent, illustating our conclusions and the effectiveness of our method. We would highly appreciate it if you could specify the reasons behind your low mark in the "evaluation" rubric, and consider improving your score. If you would like us to conduct any additional experiments that are reasonable within the rebuttal period, kindly let us know, and we will try our best to accommodate your suggestions.

---

> ### Author Response · Authors · 2024-11-15
> **Authors' Response (2/2)**
>
> Q1. The reason why pruning with DRoP seems to be less stable with a sudden average accuracy drop at density 0.3 on Waterbirds (Figure 9) is related to the strategy of reporting the average accuracy on Waterbirds as introduced by Sagawa et al. (2020). Here, instead of computing the proportion of correctly classified samples in the test set, we sum group accuracies weighted by the original unequal proportions found in the training dataset. As expected, DRoP skews the dataset composition towards minority groups that contribute much less to the average accuracy (computed in this way) than the originally overrepresented groups, which start losing in performance due to heavy pruning. Please refer to point 5 of the General Response and the corresponding revisions in the paper.
>
> We hope that we addressed all your concerns, and that you are satisfied with our responses and changes. Please consider raising your mark and recommending our paper for acceptance.
>
> **References** \
> [1] Sagawa, S., et al. "Distributionally Robust Neural Networks". ICLR 2020. \
> [2] Paul, M., et al. "Deep Learning on a Data Diet: Finding Important Examples Early in Training". NeurIPS 2021. \
> [3] Good, A., et al. "Recall Distortion in Neural Network Pruning and the Undecayed Pruning Algorithm". NeurIPS 2022. \
> [4] Ma, X., et al. "On the Tradeoff Between Robustness and Fairness". NeurIPS 2022. \
> [5] Ayed, F., Hayou, S. "Data Pruning and Neural Scaling Laws: Fundamental Limitations of Score-based Algorithms". TMLR 2023. \
> [6] Zhao, B., Bilen, H. "Dataset Condensation with Distribution Matching". ICCV 2023. \
> [7] Feng, Y., et al. "Embarrassingly Simple Dataset Distillation". ICLR 2024.

---

> > ### Comment · Reviewer_JvuS · 2024-11-17
> > **Score Raised**
> >
> > Thank you for the rebuttal.
> >
> > It answers most of my questions convincingly.
> > Yes, I agree that training large models, especially on large datasets requires a lot of computing resources.
> >
> > But given that large models are known to be more robust to distribution changes caused by image corruptions and adversarial attacks, it is not far fetched to assume that they will perform well on distribution changes due to shift in class-wise data distribution as well.
> > The proposed method not working for these models would be a perfectly acceptable and explainable limitation that can be openly discussed. Given the research community's interest in obtaining small and compressed models for real-time use of edge devices, it might become a limitation that does not play a huge role in the larger context.
> > Just, being able to see that would make the work more complete and thorough.

---

> ### Author Response · Authors · 2024-11-20
> **Authors' Reply**
>
> Thank you for such a significant increase in the rating; we are happy to see that our responses resolved your concerns. We agree with your remark regarding higher robustness of large models. We updated the manuscript with new experiments using Wide-ResNet-101 (125M parameters) on CIFAR-100 (Figure 13 and the last paragraph of Appendix D), and we believe that the results are favorable to our method.

---

### Author Response · Authors · 2024-11-15
**General Response**

We thank the reviewers for taking time to read our paper and for their valuable comments. We are happy to see the uniform acknowledgement of the importance of our work. We revised the manuscript as per your suggestions, with all changes highlighted in red in the updated file. We list these changes below.
1. The title of the paper is changed to "DRoP: Distrbutionally Robust **Data** Pruning" to avoid any confusion with model pruning, as was suggested by Reviewer JvuS.
2. As proposed by multiple reviewers, Section 2 is split into two parts to accomodate a separate "Related Work" section in order to improve the flow of the paper and make all relevant background information more accessible.
3. We added a sentence describing the issue of classification bias to Section 1 where it is first referenced.
4. As requested by Reviewer kAs6, we extend the comparison of our method to prior art in distributional robustness (page 10). In Table 2 of their paper, Pezeshki et al. (2024) report the performance of ResNet-50 on Waterbirds for a variety of existing algorithms, including JTT (Zheran Liu et al., 2021); we copy these numbers verbatim and plot them in Figure 9 to compare with Random+DRoP and other related pruning methods studied in our work. Since we are not using group attributions for validation and tuning hyperparameters for Random+DRoP, we copy the results reported for this exact setup (Pezeshki et al., 2024).
5. We provide a comprehensive explanation for the observed sudden degradation of the average accuracy of pruning with DRoP at low dataset densities in Figure 9 (ResNet-50 on Waterbirds) in the figure caption. Further, we put additional discussion and visuals about this in appendix F.
6. To provide an alternative way to examine the effectiveness of DRoP and the emerging trends, we create Table 2 (Appendix D) where we report the majority of our experimental results.

It has been pointed out by one reviewer that our evaluation lacks large-scale models. We cannot agree with this statement as we consider a variety of vision architectures including ResNet-50 trained on ImageNet from scratch. There is an abundance of highly impactful related works that were recently published in  reputable venues with empirical validation on similar or weaker models (Paul et al., 2021; Good et al., 2022; Ma et al., 2022; Ayed & Hayou, 2023; Zhao & Bilen, 2023; Feng et al., 2024), to name a few. Still, in an attempt to address this concern and reconfirm our conclusions with even larger models, we currently run additional experiments with ResNet-101 on CIFAR-100. These results are still pending and are currently not reflected in the revised manuscript. If these experiments finish before the end of the rebuttal period, we will submit another revision of the paper and post an update about it here. Note that results with larger models on ImageNet are out of reach for us: already the results we present on ImageNet with ResNet50 took over a month to run with the resources at our disposal.

In addition to submitting a revised paper, we reply to each reviewer individually to address their questions and concerns. We hope that these responses together with the revised manuscript clear up any confusion and resolve all issues that the reviewers had, and that you will consider increasing your rating for our paper.

**Referenes** \
[1] Sagawa, S., et al. "Distributionally Robust Neural Networks". ICLR 2020. \
[2] Paul, M., et al. "Deep Learning on a Data Diet: Finding Important Examples Early in Training". NeurIPS 2021. \
[3] Good, A., et al. "Recall Distortion in Neural Network Pruning and the Undecayed Pruning Algorithm". NeurIPS 2022. \
[4] Ma, X., et al. "On the Tradeoff Between Robustness and Fairness". NeurIPS 2022. \
[5] Ayed, F., Hayou, S. "Data Pruning and Neural Scaling Laws: Fundamental Limitations of Score-based Algorithms". TMLR 2023. \
[6] Zhao, B., Bilen, H. "Dataset Condensation with Distribution Matching". ICCV 2023. \
[7] Feng, Y., et al. "Embarrassingly Simple Dataset Distillation". ICLR 2024. \
[8] Pezeshki, M., et al. "Discovering Environments with XRM". ICML 2024. \
[9] Zheran Liu, E., et al.,"Just Train Twice: Improving Group Robustness without Training Group Information". ICML 2021.

---

> ### Author Response · Authors · 2024-11-20
> **Manuscript Updated**
>
> We updated the manuscipt with additional evaluations on Wide-ResNet-101 (Figure 13 and the last paragraph of Appendix D). Since this model is generally intended for large-scale datasets like ImageNet, we had to modify the original architecture slightly and remove a few downsampling layers to account for the low resolution of CIFAR-100. These experiments confirm that DRoP can be successfully used with highly overparameterized large models that are known to be more robust, as was pointed out by Reviewer JvuS.

---

### Meta-Review · Area_Chair_sTfc · 2024-12-21

**Metareview:**

The paper proposes a method, DRoP, for data pruning. The proposed approach is based on the solid theoretical and empirical analysis of previous data pruning methods that are shown to potentially result in high classification biases. DRoP is therefore phrased as distributionally robust pruning. The proposed approach is quite simple and effective and addresses an important problem of data pruning. The submitted revision is also improved in terms of presentation and clarity.  All reviewers agree that the paper is a valuable contribution to ICLR.

**Additional Comments On Reviewer Discussion:**

The paper received quite critical initial reviews. The scope of the paper as well as the experimental analysis were unclear. The rebuttal was very successful in addressing the concerns and the provided revision is much improved, which is also reflected in reviewers raising their scores (in particular, reviewer JvuS raised the score from 3 to 8 after the revision).

---

### Decision · Program_Chairs · 2025-01-22

Accept (Spotlight)